# The Polymers of Diethynylarenes—Is Selective Polymerization at One Acetylene Bond Possible? A Review

**DOI:** 10.3390/polym15051105

**Published:** 2023-02-22

**Authors:** Vyacheslav M. Misin, Irina E. Maltseva, Mark E. Kazakov, Vladimir A. Volkov

**Affiliations:** 1N.M. Emanuel Institute of Biochemical Physics, Russian Academy Science, 4 Kosygina Street, 119334 Moscow, Russia; 2SPC UVICOM Ltd., 38A Olympic Avenue, 141009 Mytischi, Russia

**Keywords:** poly-*p*-diethynylbenzene, anionic polymerization, click reaction, solid-phase polymerization, liquid-phase polymerization, catalytic activity, stereoisomers, modification, intramolecular structure

## Abstract

In this review, all available publications on the polymerization of all isomers of bifunctional diethynylarenes due to the opening of C≡C bonds were considered and analyzed. It has been shown that with the use of polymers of diethynylbenzene, heat-resistant and ablative materials, catalysts, sorbents, humidity sensors, and other materials can be obtained. Various catalytic systems and conditions of polymer synthesis are considered. For the convenience of comparison, the publications considered are grouped according to common features, including the types of initiating systems. Critical consideration is given to the features of the intramolecular structure of the synthesized polymers since it determines the entire complex of properties of this material and subsequent materials. Branched and/or insoluble polymers are formed as a result of solid-phase and liquid-phase homopolymerization. It is shown that the synthesis of a completely linear polymer was carried out for the first time by anionic polymerization. The review considers in sufficient detail publications from hard-to-reach sources, as well as publications that required a more thorough critical examination. The review does not consider the polymerization of diethynylarenes with substituted aromatic rings because of their steric restrictions; the diethynylarenes copolymers with complex intramolecular structure; and diethynylarenes polymers obtained by oxidative polycondensation.

## 1. Introduction

Polymerization of monomers containing internal and/or terminal C≡C bonds using various polymerization methods (solid phase or liquid phase) and various catalytic systems leads to polymers that have a complex of unique electrophysical, optical, and physicochemical properties. Various aspects of polymerization features and properties of substituted polyacetylenes are considered in numerous reviews [1,2,3,4,5,6,7,8,9,10,11].

At the same time, polymers synthesized by polymerization of aromatic compounds with terminal ethynyl groups H–C≡C–Ar–C≡C–H acquire their specific properties, for example, 1,4–diethynylbenzene and 1,3-diethynylbenzene [12,13,14,15,16], 1,2-diethynylbenzene [17], 1,4-diethynyl naphthalene and 4,4′-diethynyldiphenylmethane [12], 4,4′-diethynylbiphenyl [12,14,18,19], 2,5-diethynylthiophene [20], 3,6-diethynylcarbazole [21], etc. The most frequently studied monomer of this group of compounds is p-diethynylbenzene (p-DEB), also due to the fact that it is the most commercially available compound. Initially, numerous patents and articles were devoted to the creation of homo- and copolymers based on p-DEB with the formation of prepolymer resins. These prepolymers are of interest, first of all, for the creation of specific heat-resistant and ablative materials for aviation and space technology [22,23,24,25,26,27,28]. In addition, homo- and copolymers with a complex of other interesting properties have been synthesized on the basis of DEB, for example, components of solar cells, optically active polymers, polymers with photoelectric and catalytic activity, microporous and electrochromic polymers, membranes for separating gases and liquids, humidity sensors, etc. Possible applications of synthesized polymers will be given below when considering various methods of their synthesis. Many approaches and various catalytic systems have been proposed for the synthesis of these polymers.

However, it must be kept in mind that DEB is a bifunctional monomer. Therefore, its homopolymerization can proceed both along one and both –C≡CH bonds with the formation of polymers of three different types of intramolecular structure: linear, branched, and crosslinked. In turn, the resulting structure of macromolecules will fundamentally affect the properties of the polymers being formed.

The presence of –Ph–C≡CH side substituents in DEB polymers having any of the listed intramolecular structures provides an additional opportunity for the design of final polymers. Due to the targeted selection of these substituents, it is possible to change and regulate the properties of already synthesized polymers by modifying them through polymer-analogous transformations. Ethynyl and/or phenylene fragments in the side substituents –Ph–C≡C–H can play the role of chemically active fragments capable of reactions with any organic compounds, heteroatoms, or their compounds. Figure 1 shows the schemes of various modification reactions described in [29,30,31,32,33].

There are known click reactions of groups –C≡CH with various reagents. The same reactions can be recommended for modification of DEB polymers having side substituents –PhC≡CH, for example, reactions with aromatic diazides (cycloaddition azides) [34,35,36], the Sonogashira-Hagihara catalytic reaction with halogen derivatives [37,38] in accordance with the schemes in Figure 2a.

Indeed, an interesting two-stage process was demonstrated in [39]. Initially, branched copolymers of m- or p-DEB with phenylacetylene having unpolymerized groups –C≡CH were synthesized. Then, the azide-alkyne click reaction of these copolymers with azide-ended polystyrene was carried out according to the scheme shown in Figure 2b.

The –PhC≡CH substituents are present in all types of intramolecular structures of DEB polymers in varying amounts. However, only in soluble linear or branched poly-p-DEBs is it possible to carry out an effective, controlled modification reaction. In addition, only in strictly linear DEB polymers, substituents—PhC≡CH should be present in each link of the polymer. Therefore, only in linear modified polymers, in principle, it is possible to realize a structure that is a poly-conjugate chain, along which there will be sufficiently extended clusters (Figure 3). For the synthesis of such cluster systems, it is possible to use any heteroatoms or molecular fragments introduced by modification reactions listed above (Figure 1 and Figure 2a). Thus, knowledge of the intramolecular structure of DEB polymers is important information necessary to assess the possibility of modification and possible areas of their application.

To ensure that all these modification reactions are carried out with the formation of cluster chains along the main linear polyene chain, an additional necessary condition is the absence of defects or small defects in the polymer chains of the polymers used. First of all, there must be strict repeatability of polymer units with 4-ethynylphenyl substituents. In addition, it is necessary to take into account the possibility of connecting links according to the “head-to-tail” and “head-to-head” types. Therefore, the question arises about the possibility or impossibility of synthesizing a completely linear poly-p-DEB and about the structure of its probable stereoisomers.

The review considers all the methods used for the synthesis of polymers of all DEB isomers using polymerization reactions due to the C≡C bond. In necessary cases, to understand the features of the ongoing processes, the results of experiments conducted by the authors of the articles are analyzed in more detail. The probable intramolecular structure of the obtained soluble polymers, including one proposed by the authors of the reviewed articles, is considered and analyzed. The information on possible cis-trans isomers of synthesized poly-DEBs was analyzed. In addition, the proposed applications of these synthesized polymers are considered.

The review does not consider DEB copolymers, which significantly complicate the intramolecular structure of the resulting DEB copolymers. The reactions of DEB homo- and copolycondensation with the formation of ethynylene-phenylene fragments are not considered.

## 2. Synthesis, Structural Features, and Properties of p-Diethynylbenzene Polymers

### 2.1. Solid-Phase Polymerization of p-DEB Initiated by Physical Methods of Exposure

Solid-phase polymerization of p-DEB under pressure in combination with shear deformation was studied in [40]. The studies were carried out on Bridgman anvil cells, which makes it possible to obtain pressures up to 10 GPa at various temperatures. The shear stress in the sample was created by rotating one of the anvils relative to the other. It was found that at 2.5 GPa, a temperature of 22 °C, and a shear angle of 800° (2.22 complete turn), explosive polymerization occurred, accompanied by the formation of a black insoluble polymer. At a temperature of 22 °C, a shear angle of 0°, and pressures of 2.5 and 5.0 GPa, the polymer yield was 1 and 5%, respectively. The authors did not report anything about the solubility and intramolecular structure of these polymers. However, taking into account the available information about p-DEB polymers, it should be agreed that this polymer had a frequently crosslinked structure.

In article [41], it was first recorded that the crystals of p-DEB under UV or γ-irradiation in quartz cuvettes at 77 °K or at room temperature for at least 10 h acquire a yellow color. At the same time, the crystals did not change their shape, size, and surface. One of the two types of crystals formed, the darker one, showed strong dichroism. At room temperature, the irradiation was much more effective than at 77 °K. Mainly crosslinked insoluble products were formed. The soluble fraction was ≈10%. Based on optical measurements, the authors assumed that the polymer molecules formed consist of approximately 14 monomeric units. As one of the variants of the polymer chain structure, the authors proposed a linear polyene, because, in their opinion, “anisotropic absorption can be observed if predominantly one acetylene radical in each DEB molecule is involved in the polymerization reaction”. The authors consider that the reason for the termination of the photopolymerization process is the accumulation of mechanical stresses, which limit the conversion rate. However, in their opinion, the growing polymer chain does not lead to a violation of the crystal lattice of the monomer. Unfortunately, the intramolecular structure of the fractions has not been studied by any physicochemical methods, such as NMR and IR spectroscopy or the study of molecular weight distribution. The scheme of possible chain growth presented by the authors (Figure 4) is not correct, since it has an image of pentavalent carbon.

Later, a large series of articles by this group of authors was devoted to a more detailed EPR study of low-temperature (4.2–340 °K) solid-phase polymerization of crystalline p-DEB initiated by UV irradiation [42,43,44,45,46] or γ-irradiation [44,45,46,47,48]. It was found that after the termination of irradiation, the opening of the cuvettes, and the volatilization of the unreacted monomer, insoluble products remain in the main. This indirectly indicated the initial branched nature of macromolecules, although no studies of the intramolecular structure were carried out. Later, this group of authors used EPR and optical spectroscopy in the temperature range of 77–230 °K to study the generated macroradical in more detail. According to the authors, in this polymer, an unpaired electron is localized on the terminal monomeric unit of a stereoregular polymer. At 230–310 °K, its delocalization in the poly-conjugated system occurred due to the addition of a linear macroradical to the double bond of the polymer molecule. The reaction produces a branched polymer of p-DEB. Subsequently, crosslinking of polymeric branched chains with the formation of an insoluble polymer took place. The same processes occur in p-DEB, deuterated in the ethynyl group [46,47].

The same authors in [47,48,49] explain the possible reasons for the synthesis of soluble poly-p-DEB with low conversion and low molecular weight. They studied p-DEB samples by EPR after low doses of radiation (D < 500 kGy). The authors believe that at temperatures when the mobility of the polymerizing system increases (heating of irradiated samples at 77 °K; softening of the matrix during the subsequent reaction at T ≥ T_room_), the macroradical has the ability to attach to the double bonds of the formed macromolecule. This leads to the formation of short, branched polymer molecules. In addition, the macroradical can “twist”. In this case, conditions are created for the transition of an unpaired electron into the plane of conjugated π-electrons of a macromolecule, where it is delocalized along the π-conjugation chain. These reasons, according to the authors [47,48,49], lead to a loss of macroradical activity, which in turn leads to a decrease in the polymerization rate and a decrease in molecular weight.

After the first report in 1968 [41] on the radiation-induced solid-phase polymerization of p-DEB, a publication appeared [50] on the study of radiation polymerization of p-DEB and some of its derivatives. For this purpose, the authors [50] sealed p-DEB crystals in glass vacuumed (~10^−5^ torr) ampoules. The ampoules were then irradiated with γ-rays from the ^60^Co source (dose rate of 9·10^5^ rad/h) for 120 h. The brown poly-p-DEB conversion was 11%. With increasing polymerization time (125 or 150 h) an insoluble fraction appeared, and the amount of the soluble fraction decreased. The authors reported that spectral data were obtained for poly-p-DEB (^1^H NMR, ^13^C NMR, and IR spectra). However, the spectra were not given and the intramolecular structure of the polymer was not proposed.

In the article [27], an acetone-soluble prepolymer was synthesized by irradiation of solid p-DEB with ^60^Co γ-rays at a lower dose (10 kGy, 50 kGy, and 100 kGy) at room temperature. According to the results of mass spectrometry, the prepolymers required to create composites consisted only of dimers, trimers, and unreacted monomers. UV-Vis spectra show that the content of oligomers increases with an increase in the radiation dose. The authors converted these prepolymers into a crosslinked gel state by subsequent heating at 110 °C. Thus, even a branched polymer was not synthesized in the work.

In [48,49], the radical polymerization of p–DEB initiated by γ-radiation in a matrix of a glassy solution in dimethylformamide (DMF) was studied at temperatures of 77 and 300 °K. For the subsequent heating of the irradiated samples at 77 °K, in addition to the traditional heating method, microwave radiation was used. The studies were carried out by methods of EPR spectroscopy, calorimetry, size-exclusive chromatography, and gravimetry. Due to the two ethynyl groups in the monomer molecule, a crosslinked polymer is mainly formed in all experiments. In the case of using a dose of 750 kGy, a soluble polymer with a conversion rate of 4.5% was obtained. The polymer chains contained 4–8 monomeric units and had a fairly wide molecular weight distribution (${\bar{M}}_{w}/{\bar{M}}_{n}$ = 1.8, where ${\bar{M}}_{w}$ is the weight-average molecular mass, and ${\bar{M}}_{n}$ is the number-average molecular mass), which indicated a significant number of branches. At the end of the polymerization process, there was an increase in the number of closed groups –C≡CH and the regular formation of insoluble fractions. Unfortunately, the intramolecular structure of the soluble part has not been studied by spectral methods.

A low-temperature (100–210 °K) solid-phase polymerization process of p-DEB initiated by chlorine was described in [51]. For this, chlorine was frozen on a crystalline p-DEB at a temperature of 77 °K; then, the temperature was gradually raised to 210 °K at a rate of ~0.3 °K/min. The heating-cooling procedure was repeated. The polymerization process took place at noticeable rates near the melting point of chlorine and accelerated at its melting point. The unreacted monomer and chlorine were removed from the sample by vacuuming. When the procedure was repeated 10 times, a non-volatile product soluble in ethanol was obtained with a yield of ~15%_wt_. This oligomer had 2–3 units. When the procedure was repeated 20 times, a polymer fraction was also obtained, soluble only in DMF and tetrahydrofuran (THF). The synthesized polymer had 8–9 units and a very wide molecular weight distribution (${\bar{M}}_{w}/{\bar{M}}_{n}$ = 3.75). In this case, polymerization of p-DEB can occur only due to the presence of two reactive ethynyl groups, while the reaction of phenylacetylene under similar conditions led to the formation of only dimers [51]. The authors believed that branched poly-p-DEB is formed as a result of the reaction of a growing macroradical with the polyene chain of the formed polymer, as they had previously observed during UV- and γ-initiated solid-phase DEB polymerization [44,45,46,47,48,49]. Unfortunately, when discussing IR spectra, the authors did not even qualitatively analyze the areas of oscillations in the C≡CH group, let alone a quantitative analysis of the dynamics of changes in these groups. In all cases, all fractions contained chlorine in an amount of ~40–50%_wt_. The IR spectra showed bands at 738, 786, and 878 cm^−1^ characteristic of the C–Cl bond, but the authors did not indicate possible places of chlorine addition.

The mechanism of polymerization of acetylenes by molecular chlorine at low temperatures is considered in more detail in [52,53]. It is shown that without external energy influence, radicals are spontaneously formed. The formed radicals initiate polymerization and chlorination reactions of p-DEB, which occur when prepared at 77 °K mixtures are heated. The features of the intramolecular structure are not discussed in any way, and when discussing IR spectra, the characteristic bands of C≡C and C–H bonds in the –C≡C–H group are not even mentioned.

The depositions of p-DEB in an ultrahigh vacuum on Cu(111) substrates and subsequent annealing at temperatures from 175 to 350 °C were studied in [54]. Annealing of self-assembled DEB structures led to surface covalent aggregates and meshes that did not have a linear chain structure.

### 2.2. Gas-Phase Polymerization of p-DEB

The process of frontal polymerization of p-DEB occurring as a result of high-temperature initiation at elevated nitrogen pressure (P = 2–6 MPa) in a standard Crawford bomb has been studied in [55]. Thermal initiation of the process was carried out using a nichrome coil when an alternating current of 8 V was applied to it. According to the authors [55], after that, the process of intense evaporation began, followed by rapid polymerization of p-DEB in the gas phase. Due to the high thermal effect of polymerization, the temperature in the reactor remains high even after the thermal initiation impulse is switched off. The p-DEB has a low melting and boiling points, as well as high volatility. Therefore, after evaporation and polymerization of the first upper layer, each subsequent layer of p-DEB also evaporated and polymerized in the gas phase. As a result, an insoluble crosslinked polymer in the form of a black cylindrical rod was obtained. As a result of the Raman spectrum analysis, the authors assumed that the polymer had a graphite structure. Using SEM, microspheres were imaged with a rather narrow size distribution and with an average diameter of 100–200 nm. It is not excluded that in this case, condensation of p-DEB vapors into the liquid phase and subsequent liquid-phase polymerization in the melt at a high temperature may occur, similar to [56].

### 2.3. Liquid-Phase Polymerization of p-DEB without the Use of Catalysts

Liquid-phase polymerization is the method that, in principle, makes it possible to obtain poly-p-DEB in sufficient quantities and a controlled structure to create the required final materials.

Heating (100–280 °C) in sealed ampoules in an argon medium of mixtures of p-DEB and polynaphthalene, anthracene, and polynaphthalene copolymer with benzene [57] and with polyphenylene (1/2 wt.) [58] led to the formation of either soluble or insoluble products. According to the authors (in accordance with the IR spectra), depending on the experimental conditions, p-DEB was grafted to polyarylenes (soluble products were obtained) or polyarylenes were crosslinked with grafted p-DEB with the formation of trans-vinyl bridges.

The possibility of using p-DEB as a dispersant of solid fuels in a ramjet engine is considered in [59,60]. For this, the kinetics of heat release during its thermally initiated polymerization under isothermal conditions in the temperature range of 90–150 °C-steam in a Crawford bomb at an initial pressure of 2 MPa has been studied.

Apparently, the first attempt at liquid-phase homopolymerization of p-DEB in a melt was described in [56,61] in 1967. The authors polymerized DEB in sealed ampoules in a melt under argon for 6 h at temperatures of 100–185 °C, always obtaining a dark insoluble product. The intramolecular structure of polymers has not been investigated.

The thermal curing reaction of p-DEB was studied in [62] by comparing the activation energies of the thermal curing reactions of various acetylene monomers. p-DEB as one of the objects of comparison is considered together with other acetylene monomers: o- and m-DEB, 1,2,4- and 1,3,5-triethynylbenzenes, as well as 1,2,4,5-tetraethynylbenzene. The aim of the work was to obtain complete information about the features of the curing processes of individual phenylacetylene monomers and their various compositions in the synthesis of resins with terminal acetylene groups (PAA resins). This work did not consider the structural features of the products obtained.

Interaction in a sealed degassed ampoule of p-DEB with iodine (DEB/iodine = 1/1 mol) at 130 °C for 6 h produced a yellow polymer containing 60–68% iodine and soluble in benzene and chloroform [63]. A solid insoluble polymer block was obtained at a ratio of p-DEB/iodine ≤ 1/0.7 mol. Increasing the heating time also resulted in the formation of an insoluble polymer.

The authors believe that the polymerization mechanism first involves the process of halogenation, and then dehalopolycondensation in accordance with the scheme (Figure 5) on the example of polymerization of a structural analog—phenylacetylene.

Based on IR (including strong bond ν_C–I_ = 595 cm^−1^) and ^1^H-NMR spectra, the authors propose a branched structure of soluble poly-p-DEB (Figure 6a).

Films made of soluble poly-p-DEB had σ_20_ = 10^−10^ Ω^−1^cm^−1^. After heat treatment at 130 °C, the polymer became insoluble. The iodine content decreased to 27%, and the absorption bands characteristic of C–I and C≡C bonds disappeared. The authors explained this by the process of disappearance of links with substituents –CI=CHI in accordance with the scheme shown in Figure 6b.

After boiling a 15% solution of p-DEB in DMF (b.p.= 153 °C) in a nitrogen atmosphere for 48 h, a polymer with a conversion rate of 29% and ${\bar{M}}_{n}$ = 1459 was obtained [64]. An increase in the time of thermal polymerization to 49 h led to the formation of a gel. Based on the results of IR and NMR spectra, the authors believed that the polymer had a branched structure. The curing of poly-p-DEB was studied using dynamic and isothermal differential scanning calorimetry (DSC).

In the case of irradiation with UV light or γ-60Co (≈2 Mrad) by radiation of p-DEB solutions in ethanol (10 mmol/mol) at 77 and 300 °K, the samples, respectively, remained colorless or slightly yellow. This indicated the absence of the polymerization process of the monomer in solution under the influence of the radiation used [41,45].

Taking into account the reaction schemes given in [65], p-DEB was used in this work to create a prepolymer preliminarily. The method of creating a prepolymer is not given in full. The process of gel formation (temperature 100 °C) is unclear in the absence of any solvent since it is known that pyrolysis of p-DEB at this temperature leads to an insoluble solid product [56,61,62]. Subsequently, the prepolymer was used to create compositions with fir powder. The structure of the prepolymer is not investigated or discussed by the authors. The reaction schemes suggest the probable structure of the crosslinked product. However, these structural elements are most likely characteristic of polymers based on o-DEB and m-DEB.

### 2.4. Liquid-Phase Polymerization of p-DEB with the Use of Catalysts

The first attempts at a liquid-phase synthesis of poly-p-DEB in the presence of catalytic systems are described in the patent [66] and in articles [67,68,69,70,71]. It was reported in [68,69] that in the presence of the iso-(C_4_H_9_)_3_Al—TiCl_4_ complex (Al/Ti = 0.5), an insoluble polymer is formed in 6 h with a yield of 40%. In [70], they only state the fact of the formation of an insoluble polymer in the presence of iso-(C_4_H_9_)_3_Al and TiCl_4_ without specifying any polymerization conditions. Nevertheless, the authors [70,72] believed that 1,2,4- or 1,3,5-cyclotrimerization of p-DEB occurs first, as was observed in the case of phenylacetylene [73]. Subsequently, the polymer mesh is formed according to the scheme (Figure 7).

The patent [66] reported on the polycyclotrimerization reaction of diethynylarylenes using catalysts ${\left[{\left(\mathrm{RO}\right)}_{3}P\right]}_{n}\cdot \mathrm{CoHal}$, (where R = Alk_C≤6_, Hal = Cl, Br, I). The goal is the synthesis of thermosetting polymers cured at 150–200 °C. However, only one example describes the synthesis of poly-p-DEB only in the presence of triethyl phosphate cobalt iodide complex ${\left[{\left(\mathrm{EtO}\right)}_{3}P\right]}_{4}\cdot \mathrm{CoI}$. The yield of the insoluble polymer was 70.6% for 1 h in boiling ethanol. The IR spectrum of the crosslinked polyphenyl product shows the presence of trisubstituted benzene nuclei in the polymer molecule (at 810–850 cm^−1^).

The studies [67,74] report on the continuation of studies on the homopolymerization of diethynylarylenes, including p-DEB, in the presence of various complex cobalt catalysts. Apparently, the catalyst was the ${\left[{\left(\mathrm{EtO}\right)}_{3}P\right]}_{4}\cdot \mathrm{CoI}$ complex mentioned in [66]. According to the authors, highly crosslinked polyphenylenes are formed by the mechanism of polycyclotrimerization with the formation of 1,3,5- and 1,2,4 (98%)-substituted phenylene fragments (Figure 8). At the same time, ≡CH proton signals almost completely disappear in the ^1^H-NMR spectra.

The article [75] reports on the approbation of a larger number of the same trialkylphosphate complexes of cobalt. However, the kinetic features of p-DEB polymerization are considered in the presence of only one trialkylphosphite complex of cobalt ${\left[{\left(C_{2}H_{5}O\right)}_{3}P\right]}_{4}\cdot \mathrm{CoBr}$. The reaction was carried out in a solution of benzene, toluene, dioxane, or alcohol in an inert gas medium at temperatures of 50–100 °C. The total yield of non-soluble products varied in the range of 15–68%, and the total yield of soluble products was 20–44%. The molecular weight of soluble branched polymers of p-DEB reached 2600. NMR spectra showed that during polymerization, the number of ethynyl group protons gradually decreases so that by the end of the reaction, the ratio of the total number of olefin and aromatic protons to ethynyl protons is $H_{\mathrm{ol}+\mathrm{ar}}/H_{\mathrm{eth}}=8/$1. This indicated the presence of branches in the polymer. The authors proposed a scheme for the polycyclotrimerization of p-DEB through the formation of a complex of three ethynyl groups with a transition metal atom (Figure 9), although they also allowed for other intermediate stages.

The structure of highly branched polyphenylene for poly-p-DEB (Figure 9) synthesized using complexes ${\left[{\left(\mathrm{RO}\right)}_{3}P\right]}_{n}\cdot \mathrm{CoHal}$ and proposed in [66,67,74,75], was confirmed in [76], who studied the thermal characteristics of poly-p-DEB, and in a series of reviews [71,77,78,79].

A large number of publications have been devoted to the polymerization of p-DEB in the presence of the nickel acetylacetonate/triphenylphosphine $\mathrm{Ni}{\left(C_{5}H_{7}O_{2}\right)}_{2}\cdot {\mathrm{Ph}}_{3}P$ complex. In some examples of early patents [80,81,82,83], the synthesis of poly-p-DEB in N_2_ medium in a solution of anhydrous dioxane with a conversion of 57% was reported. The polymer is recommended to be used in compositions when creating carbon materials. Poly-p-DEB had ${\bar{M}}_{n}$ = 2900. For synthesis, 71 parts of p-DEB, 1.062 parts of $\mathrm{Ni}{\left(C_{5}H_{7}O_{2}\right)}_{2}$, 2.124 parts of Ph_3_P and 737 parts of anhydrous dioxane were loaded into the reactor. The authors of this review recalculated the mass fractions into molars in order to compare these indicators with those indicated in subsequent publications. The ratio of the values of p-DEB/Ni(Acac)/Ph_3_P = 137/1/2 was obtained. According to the NMR spectrum (the spectrum is not shown), the branched polymer molecules had phenylene fragments. The prepolymer contained 15.0% acetylene groups. A significant excess of the number of aromatic protons compared to the number of acetylene protons in poly-p-DEB indicated the formation of such protons due to the effective polycyclotrimerization reaction of ethynyl groups in the presence of a $\mathrm{Ni}{\left(C_{5}H_{7}O_{2}\right)}_{2}\cdot {\mathrm{Ph}}_{3}P$ catalyst. However, the phrase of the authors (“Analysis by NMR as described above showed the prepolymer to have a ratio of aromatic protons to olefinic protons of greater than 30/1”) is surprising, because in PMR spectra these protons are not distinguishable. The authors probably meant acetylene protons, not olefin protons.

A group of authors in later publications [84,85,86,87,88] studied the polymerization of p-DEB in the presence of the same $\mathrm{Ni}{\left(C_{5}H_{7}O_{2}\right)}_{2}\cdot {\mathrm{Ph}}_{3}P$ catalyst under various conditions: component ratio, solvent type, temperature, and effect of LiCl addition. The purpose of the research is to develop conditions for the synthesis of poly-p-DEB, capable of forming heat-resistant and easily carbonized fibers from it. The process of poly-p-DEB synthesis was carried out in a medium of dry polar solvents (DMF, DMA, or N-MP) at temperatures of 70–120 °C for 1–9 h in an inert gas medium. The conversion reached 96%. The polymer solution was subjected to preliminary structuring by heating it at 100 °C for 6 h. Then it was molded into an aqueous precipitation bath through a spinneret having 300 holes with a diameter of 0.05 mm. The fiber dried at 50 °C was subjected to heat treatment up to 250 °C at a heating rate of 5 °C/min. As a result, the authors obtained an elementary fiber with a diameter of 0.025 mm and a strength of 21–27 kg/mm^2^.

The patent [84] shows the only used ratio of the components of the catalytic complex ${\mathrm{Ph}}_{3}P/\mathrm{Ni}{\left(C_{5}H_{7}O_{2}\right)}_{2}$ = 1/2. A later patent [85] indicated a different ratio of the components of the reaction system, which was changed in the intervals of DEB/${\mathrm{Ph}}_{3}P/\mathrm{Ni}{\left(C_{5}H_{7}O_{2}\right)}_{2}$ = 1.4–1.6/0.1–0.12/0.005–0.006 mol/L. An increase in the conversion of p-DEB was noted [87] due to an increase in the concentration of Ph_3_P in the reaction medium and in the case of the addition of LiCl. The polymer yield was quantitative at a ratio of DEB/${\mathrm{Ph}}_{3}P/\mathrm{Ni}{\left(C_{5}H_{7}O_{2}\right)}_{2}$ = 313/1/20 mol/L. According to the authors [86], polymerization occurred with the formation of an intermediate active center—hydride acetylenide (Figure 10a)—and therefore poly-p-DEB has a linear structure of non-branched polyene (Figure 10b). The authors substantiated this assumption by comparing the intensities of the bands of phenyl (1510 cm^−1^) and ethynyl (3300 cm^−1^) groups in the IR spectra of p-DEB and poly-p-DEB [87].

Unfortunately, in this article, the corresponding IR spectra were not given and the NMR spectra of poly-p-DEB were not taken, which would make it possible to determine the intramolecular structure of the polymer. Later, these authors reported in an article [30] that under the previously used [85,87] synthesis conditions, polymers are formed that “have a predominantly linear structure with a trans-transoid conformation and contain free ethynyl groups on almost every aromatic nucleus”. That is, in fact, the synthesized polymer had a certain number of branches in the main chain.

Thus, the intramolecular structure of poly-p-DEB proposed in [30,87] differs from the structure proposed in [80,83]. A possible reason may be the difference in the conditions of the p-DEB polymerization reaction. Indeed, our recalculation of the mass loading of components indicated in [80,83] for loading in the molar ratio gave the following values: ${\mathrm{Ph}}_{3}P/\mathrm{Ni}{\left(C_{5}H_{7}O_{2}\right)}_{2}$ = 2/1. On the other hand, in [85] a different ratio of these components was used during polymerization, as indicated above. Unfortunately, [87] it is not indicated under what specific conditions poly-p-DEB was obtained, the IR spectrum of which was studied.

In an article that appeared much later [64], the authors argued that when using the same complex $\mathrm{Ni}{\left(C_{5}H_{7}O_{2}\right)}_{2}\cdot {\mathrm{Ph}}_{3}P$ catalyst, the main reaction should be polycyclotrimerization of p-DEB, proceeding according to the scheme in Figure 11. However, in the polymerization scheme (Figure 11), the authors, probably mistakenly, depicted a cyclotrimer obtained from m-DEB and not from the original p-DEB.

The authors of this article [64] allowed a small part of the reactions to occur with the formation of linear and/or branched structures. As a result of the reaction, a soluble prepolymer was formed for 3 h with a yield of 79% and a value of ${\bar{M}}_{n}=$ 1875. It was shown that the Ni-catalyzed polymerization of diethynylbenzene is an effective way to synthesize prepolymers needed to produce composites. An increase in the polymerization time to 3.5 h led to the formation of a gel. The FT-IR spectra of soluble catalytic and thermal prepolymers showed characteristic bond vibrations of acetylene groups at 3300 cm^−1^ (stretching ≡C–H) and 2106 cm^−1^ (stretching C≡CH), respectively. The ^1^H-NMR spectra show the same signals, which differ in intensity for some protons, including acetylene protons (≡C–H) at 3.1–3.3 ppm. The authors believed that the spectra indicate the realization of a branched structure with the preservation of part of the groups –C≡CH. It is this structure that ensures the formation of gels when certain polymerization times are reached. Unfortunately, the absence of integral ^1^H-NMR spectra did not allow us to confirm this ratio of different types of units. The purpose carried out by the authors’ research is to study the curing process of poly-p-DEB, which is a good resin for creating carbon matrices. Using dynamic and isothermic DSC, it was found that the exothermic crosslinking process began at 120 °C, reached a maximum at 210 °C, and ended at 300 °C. More than 85% of –C≡CH groups reacted during curing. For comparison, thermal poly-p-DEB synthesized by boiling a solution of p-DEB in DMF was studied. It was found that the coke residue of catalytic poly-p-DEB (pyrolysis at 800 °C) was 79–86%. On the contrary, the coke residue of thermal poly-p-DEB was only 74%, which the authors explained by the presence of structural defects in polymer chains.

In [89], a soluble prepolymer from p-DEB was synthesized using 13, 16 and 19%_wt_. of the same catalyst $\mathrm{Ni}{\left(C_{5}H_{7}O_{2}\right)}_{2}\cdot {\mathrm{Ph}}_{3}P$ according to the method taken from [64]. The conversion of the soluble polymer was 77% for 2 h (19%_wt_. of the catalyst), without reaching the gelation region. The prepolymer was evaluated as a polymeric precursor of monolithic vitreous carbon. Based on ^1^H-NMR spectra, the authors confirmed the presence of a branched polyphenylene structure. However, in the polymerization scheme (Figure 12), taken from [64], the authors also mistakenly depicted a cyclotrimer obtained from three m-DEB molecules.

Indeed, the resin synthesized from a mixture of p- and m-DEP in the presence of $\mathrm{Ni}{\left(C_{5}H_{7}O_{2}\right)}_{2}\cdot {\mathrm{Ph}}_{3}P$ or ${\mathrm{TiCl}}_{4}\cdot {\mathrm{Et}}_{2}\mathrm{AlCl}$ complexes is also formed due to the cyclotrimerization process (Figure 13) [23]. In addition, the cyclotrimerization of acetylenes in the presence of $\mathrm{Ni}{\left(C_{5}H_{7}O_{2}\right)}_{2}\cdot {\mathrm{Ph}}_{3}P$ is indicated by the authors [24] who studied the co-cyclotrimerization of mixtures of DEB and ethylphenylacetylene isomers and proposed the branched cyclotrimer structure.

During polymerization of p-DEB and other diethynylaromatic monomers in the presence of rhodium catalysts [Rh(cod)ac] and [Rh(nbd)acac] (cod: cycloocta-1,5-diene; nbd: norborna-2,5-diene; acac: acetylacetone) in the medium of CH_2_Cl_2_, the authors [14] synthesized insoluble microporous polymers with yields of 80 and 85%, respectively. The polymers had S_BET_ values of 512 and 809 m^2^ g^−1^, respectively, and a micropore volume of 0.160 and 0.247 cm^3^ g^−1^. The adsorption of H_2_ on polymers was reversible, in contrast to nitrogen adsorption. This allowed the authors to consider it possible to optimize the physical properties of polymers in order to create materials for gas storage. Using the SEC and ^13^C CP/MAS NMR spectroscopy, it was found that the polymer consisted of polyene conjugated meshes with ethynylarylene substituents and crosslinked with arylene linkers (Figure 14).

The soluble polymer was obtained only in the presence of a [Rh(nbd)acac] catalyst after 2–3 min (no more!) after the start of synthesis. The authors found that this poly-DEB was highly branched or partially crosslinked even at the beginning of its formation.

In a later article [15], the authors investigated the effect of reaction conditions (type of catalyst, solvent, temperature, concentration of catalyst, and monomer) on the resulting meshes of poly-p-DEB and the adsorption capacity of synthesized polymers. The use of [Rh(nbd)acac] and [Rh(nbd)Cl]_2_/Et_3_N complexes in CH_2_Cl_2_ medium provided the synthesis of insoluble, non-swellable crosslinked microporous polymers for 3 h at room temperature with yields of 85 and 77%, respectively. A red precipitate fell out immediately after mixing the reagents, or after a few minutes. An insignificant amount of unreacted –C≡C–H groups remained in the polymers (Figure 15). During the polymerization of p-DEB in the presence of [Rh(nbd)acac], the S_BET_ value of the obtained poly-p-DEB increased depending on the solvent used in the series: THF << pentane < benzene < methanol < CH_2_Cl_2_.

The S_BET_ value increased with increasing reaction time and temperature, as well as with an increase in the initial concentration of the monomer. The maximum value of S_BET_ = 1469 m^2^g^−1^ was obtained for poly-p-DEB synthesized using [Rh(nbd)ac ac] in CH_2_Cl_2_ for 72 h at 75 °C. The diameter of the micropores was about 1 nm. In addition to micropores, poly-p-DEB contained mesopores. The heat treatment of poly-p-DEB at 280 °C caused the complete disappearance of the –C≡CH side groups with the formation of new crosslinking. The heat-treated poly-p-DEB showed a higher adsorption capacity for H_2_ and CO_2_ compared to the non-thermalized polymer. At the same time, metathesis catalysts (WCl_6_/Ph_4_Sn, MoCl_5_/Ph_4_Sn, Mo-carbene) proved to be ineffective in polymerization in benzene solution at room temperature and reaction time up to 24 h, demonstrating a conversion of 39, 2, and 11%, respectively.

The ability of numerous low molecular weight acetylene compounds (including 1,3- and 1,4-diethynylbenzenes) to catalyze model acetylation and esterification reactions was analyzed in [90]. The authors argued that acidic hydrogen of ethynyl groups –C≡CH is capable of carrying out acetylation and esterification reactions, usually catalyzed by acid. The logical development of these studies is the study of the catalytic activity of polymers having –C≡CH groups [91]. In the article, the authors [91] for the first time proved the principal possibility of the catalytic activity of this insoluble, non-swellable, super-crosslinked red poly-p-DEB with a constant micro/mesoporous structure and a specific surface area S_BET_ up to 1007 m^2^g^−1^. The synthesis was carried out in the presence of a catalyst [Rh(nbd)acac] in accordance with the methodology [15]. Part of the groups –C≡CH was not polymerized and remained free (0.4 per monomer unit). Due to the weakly acidic hydrogen atoms of these unreacted ethynyl groups, the polymer should have been active in acid-catalyzed reactions. The authors evaluated the heterogeneous catalytic activity of the acidic acetylene hydrogen using the acetylation reaction by methanol of five aldehydes and two ketones at 60 °C. The conversion was 0.5–48% for 0.25 h and 14–49% for 7 h. The control experiment showed that polyphenylacetylene, a structural analog of poly-p-DEB, that does not have groups –C≡CH, did not activate this reaction. On the other hand, monomeric phenylacetylene catalyzed a homogeneous acetylation reaction of isopentanal with methanol with a conversion of 8%. However, the catalytic activity of phenylacetylene was naturally lower than that of poly-DEB (32%). This was explained by the lower acidity of the acetylene hydrogen atom of phenylacetylene, compared with the acidity of the acetylene hydrogen atoms present in the side substituents –PhC≡CH conjugated poly-p-DEB.

In the review devoted to the use of rhodium catalysts for the polymerization of various mono- and diacetylenes, including p-DEB [9], the main types of Rh-based catalysts, including those immobilized on both organic and silicon carriers, were considered. The review indicates that during the polymerization of various diethynylarenes, strongly crosslinked polyacetylene meshes were formed.

In the presence of a complex catalyst $\mathrm{Ni}{\left({\mathrm{PPh}}_{3}\right)}_{2}{\left(C\equiv C–C_{6}H_{4}–C\equiv \mathrm{CH}\right)}_{2}$ in a mixture of solvents dioxane-toluene (1/1 by volume) in an atmosphere of N_2_ at 25 °C for 6 h with a conversion of 70%, soluble poly-p-DEB having ${\bar{M}}_{w}$ = 3000–30,000 was obtained [92]. The synthesized polymers had low electronic conductivity. However, in the presence of ambient humidity in tablet samples doped with chloric or sulfuric acids, the conductivity of the samples sharply increased due to proton migration through the material. The authors suggested that the mechanism of charge transfer through materials is based on the proton conductivity of a thin layer of an aqueous acid solution adhering to the polymer grains. It was proposed to use such materials in the development of humidity sensors since the doped polymers showed a reversible change in conductivity depending on humidity. In [93], the same polymer was studied as in [92], but in the form of a film sample applied to a surface acoustic wave (SAW) delay line oscillator. The polymer structure was not given. Poly-p-DEB has shown properties that made it possible to recommend it as a good moisture-sensitive material. The effect of the dopant concentration (HClO_4_) on the electrical properties and relative humidity of this poly-p-DEB was studied in [94]. The doped poly-p-DEB was characterized using electron spin resonance (ESR), UV-vis, IR, X-ray photoelectron spectroscopy (XPS), and scanning electron microscopy (SEM). The conductivity of the doped polymer increased by 11 orders of magnitude, reached 10^3^ S·cm^−1,^ and increased with an exponential trend depending on relative humidity (below 60%). The authors believed that the formation of a complex with charge transfer of poly-p-DEB with the hydroxonium ion H_3_O^+^ took place. This led to the delocalization of the charge of the π-electron along the polymer chain and a decrease in the transition energy of π→π*. A rectilinear dependence of the current (four orders of magnitude) on the relative humidity (10–60%) on a logarithmic scale was found in the doped poly-p-DEB. The authors of all these publications declared the trans-linear structure of the polymer in the absence of any instrumental studies of the intramolecular structure.

This group of authors in the following article [95] continued the research of poly-p-DEB (catalyst $\mathrm{Ni}{\left({\mathrm{PPh}}_{3}\right)}_{2}{\left(C\equiv C–C_{6}H_{4}–C\equiv \mathrm{CH}\right)}_{2}$) as a relative humidity (RH%) sensor. Electrical measurements were carried out on gold electrodes. Thin-film coatings for them were prepared in three ways: (A) a Langmuir-Blodgett (LB) monomolecular layer deposition, (B) spin coating, and (C) transferring PDEB film formed on the water surface. All sensors had a low impedance (in the range of 10^3^–10^7^ Ω in 15–92% RH) and a small hysteresis (Figure 16). However, the sensor manufactured by the LB deposition method had a better response. The authors explained this by the smallest thickness and more ordered structure of the membrane. Using ^1^H NMR spectroscopy, the authors introduced certainty to the knowledge of the intramolecular structure of the synthesized poly-p-DEB. They attributed a relatively strong signal at 6.85 ppm to trans-polyene protons. The ratio of the sum of olefin and aromatic protons to ethynyl protons was $H_{\mathrm{ol}+\mathrm{ar}}/H_{\mathrm{eth}}=8/1$ instead of 5/1 for the linear structure. The authors believed that the polymer had a trans-polyene structure with a certain number of side groups, probably in the branched structure in Figure 10c.

In [96], the optical, photoluminescent, and electroluminescent properties of poly-p-DEB synthesized in the presence of the same $\mathrm{Ni}{\left({\mathrm{PPh}}_{3}\right)}_{2}{\left(C\equiv C–C_{6}H_{4}–C\equiv \mathrm{CH}\right)}_{2}$ catalyst and under the conditions specified in [92,93,94] were investigated. Accordingly, the authors [96] declared the linear structure in Figure 10b previously proposed in [94], contrary to the information available in [95] about the presence of a branched structure in Figure 10c. For poly-p-DEB, strong photoluminescence was detected when illuminated by an ultraviolet lamp and experimental single-layer LEDs were manufactured using poly-p-DEB as an emitting layer [96]. This allowed the authors to recommend the polymer as a luminescent material for light-emitting devices. From the results of comparing the absorption spectra of poly-p-DEB, its two derivatives, and polyphenylacetylene, the authors concluded that the aryl group in the side chain has a significant effect on the absorption spectra of poly (phenylacetylene) derivatives. However, this conclusion is in contradiction with the results of the article [57], in which the hyperfine structure of the EPR spectra of nitropolyarylvinylenes, including poly-4-nitrophenyl acetylene, was investigated. The authors believe that there is no interaction between the electrons of the main chain and the pendants in the polymer. Thus, the possible reasons for the spectral differences of polymers synthesized in [96] can be different lengths of polymer chains, different cis- or trans-isomerization, as well as the presence of phenylene fragments in the chain. The conclusions in [96] will be valid provided that the same structure of the main polymer chains of the studied polymers is proved, which was not proven in this work.

In [97], photoluminescence, electroluminescence, and conductivity of doped (HClO_4_, H_2_SO_4_, HBF_4_, I_2_, FeCl_3_, CH_3_COOH) poly-p-DEB synthesized in the presence of $\mathrm{Ni}{\left({\mathrm{PPh}}_{3}\right)}_{2}{\left(C\equiv C–C_{6}H_{4}–C\equiv \mathrm{CH}\right)}_{2}$ catalyst and under the conditions specified in [92,93,94] were investigated. However, according to the authors of this article, the polymer has a branched trans-structure (Figure 10c) with ${\bar{M}}_{w}$~11,000 and a polydispersity index of 3.0. It was found that the use of HClO_4_ made it possible to increase the initial conductivity of poly-p-DEB (10^−14^–10^−13^ S·cm^−1^) by 10 orders of magnitude to 6.3·10^−3^ S·cm^−1^. Other dopants (H_2_SO_4_, CCl_3_COOH, HBF_4_, I_2_, FeCl_3_, p-CH_3_-C_6_H_4_-COOH, CH_3_COOH) showed lesser effect.

The same opinion about the existence of a branched poly-p-DEB structure is shared by the authors [98] who study the dependence of the conversion, ${\bar{M}}_{w}$ and ${\bar{M}}_{w}/{\bar{M}}_{n}$ on various factors: the concentration of the monomer and the catalyst $\mathrm{Ni}{\left({\mathrm{PPh}}_{3}\right)}_{2}{\left(C\equiv C–C_{6}H_{4}–C\equiv \mathrm{CH}\right)}_{2}$ (Figure 17 and Figure 18), temperature (0–50 °C), gas medium (N_2_, CO_2_, H_2_, air), solvent type (1,4-dioxane, DMSO, THF, toluene, benzene, CH_2_Cl_2_ etc.), and additives (H_2_O, EtOH, Et_3_N, Ph_3_N). The authors showed that a trans polymer with ${\bar{M}}_{w}$ up to 27,100 has a ratio of $H_{\mathrm{ol}+\mathrm{ar}}/H_{\mathrm{eth}}=8/1$, indicating branched chains.

It was shown in [99] that not only $\mathrm{Ni}{\left({\mathrm{PPh}}_{3}\right)}_{2}{\left(C\equiv C–C_{6}H_{4}–C\equiv \mathrm{CH}\right)}_{2}$, but also other Ni complexes, including those containing polar acetylides, have catalytic activity with respect to p-DEB (although to a lesser extent). Air-stable complexes $\mathrm{Ni}{\left({\mathrm{PPh}}_{3}\right)}_{2}{\left(C\equiv C–{\mathrm{CH}}_{2}{\mathrm{OCOCH}}_{3}\right)}_{2}$, $\mathrm{Ni}{\left({\mathrm{PBu}}_{3}\right)}_{2}{\left(C\equiv C–C_{6}H_{4}–C\equiv \mathrm{CH}\right)}_{2}$, $\mathrm{Ni}{\left({\mathrm{PBu}}_{3}\right)}_{2}{\left(C\equiv C–{\mathrm{CH}}_{2}{\mathrm{OCOCH}}_{3}\right)}_{2}$ in a mixture of 1,4-dioxane/toluene (1/1 by volume) were provided the yield of soluble poly-p-DEB 58, 28, and 52% for 10 h (at 25 °C), 48 (60 °C) and 33 (60 °C) hours, respectively. The polymers had ${\bar{M}}_{w}\;$= 13,000–28,000. Complexes containing PBu_3_ ligands showed lower activity than complexes containing PPh_3_ ligands. The authors attributed this to the higher basicity and interconnected donor capacity of PBu_3_, which increased the electron density of the metal–carbon bond and reduced the reactivity of this bond. The replacement of Ni by Pd in one of the complexes with the same ligands led to a decrease in the activity of the complex. For the $\mathrm{Ni}{\left({\mathrm{PPh}}_{3}\right)}_{2}{\left(C\equiv C–C_{6}H_{4}–C\equiv \mathrm{CH}\right)}_{2}$ complex, the conversion of 74.4% was achieved at a temperature of 25 °C for 3 h, while for the $\mathrm{Pd}{\left({\mathrm{PPh}}_{3}\right)}_{2}{\left(C\equiv C–C_{6}H_{4}–C\equiv \mathrm{CH}\right)}_{2}$ complex the conversion of 79.9% was achieved at a higher temperature of 60 °C in only 18 h. FT-IR, ^1^H, and ^13^C NMR spectroscopy confirmed the branched π-conjugate structure of the polyene chain poly-p-DEB. According to the authors, an additional confirmation is a wide and multimodal molecular mass distribution (${\bar{M}}_{w}/{\bar{M}}_{n}$ up to 3.0) and small Mark–Houwink constants (a = 0.26).

In [100], the catalytic activity of complex catalysts containing various phosphine and alkynyl ligands was compared: NiL_2_(C≡CR)_2_; NiL_2_(C≡CR)Cl and NiL_2_Cl_2_, where L_2_ = (PPh_3_)_2_, (PBu_3_)_2_ and Ph_2_PCH_2_CH_2_PPh_2_; R = p-C_6_H_4_C≡CH, C_6_H_5_, H, CH_2_OH, and CH_2_OCOCH_3_. Soluble π-conjugated poly-p-DEB with yields up to 95% and ${\bar{M}}_{w}$ up to 48,000 was synthesized in an atmosphere of N_2_ in a mixture of dioxane/toluene at 25 °C for 3 h. It was found that the activity of nickel acetylides with various phosphine ligands decreased in the sequence: $\mathrm{Ni}{\left({\mathrm{PPh}}_{3}\right)}_{2}{\left(C\equiv C–C_{6}H_{4}–C\equiv \mathrm{CH}\right)}_{2}$ > $\mathrm{Ni}{\left({\mathrm{PBu}}_{3}\right)}_{2}{\left(C\equiv C–C_{6}H_{4}–C\equiv \mathrm{CH}\right)}_{2}$ > $\mathrm{Ni}{\left({\mathrm{PPh}}_{3}\right)}_{2}({\mathrm{Ph}}_{2}{\mathrm{PCH}}_{2}{\mathrm{CH}}_{2}{\mathrm{PPh}}_{2}){\left(C\equiv C–C_{6}H_{4}–C\equiv \mathrm{CH}\right)}_{2}$. The activity of nickel acetylides containing (PPh_3_)_2_ and nonpolar alkynyl ligands is slightly higher than the activity with the polar ligand C≡CCH_2_OCOCH_3_. In the case of nickel acetylides containing (PBu_3_)_2_, an inverse dependence was generally observed. The $\mathrm{Ni}{\left({\mathrm{PPh}}_{3}\right)}_{2}{\mathrm{Cl}}_{2}$ complex did not show catalytic activity. The work did not consider the structure of the resulting poly-p-DEB and did not carry out the necessary spectral studies for this.

In [101], the mechanism of polymerization of p-DEB in the presence of $\mathrm{Ni}{\left({\mathrm{PPh}}_{3}\right)}_{2}{\left(C\equiv C–C_{6}H_{4}–C\equiv \mathrm{CH}\right)}_{2}$ was investigated in a mixture of DO/toluene at 60 °C for 48 h. It was shown that additions of the electron donors hydroquinone, NEt_3_, EtOH, and H_2_O (electron donor/catalyst = 5 mole ratio) affected the conversion of the monomer, varying in the range of 54–88%. The values of ${\bar{M}}_{w}$ and ${\bar{M}}_{w}/{\bar{M}}_{n}$ changed, respectively, in the intervals 1.0–2.7·10^−4^ and 1.6–8.6. The authors proposed a coordination insertion mechanism by the insertion of monomers into metal–carbon σ-bonds. According to the authors, for a large-sized p-DEB, trans-connection is most appropriate. According to the authors, the chain transfer to the monomer probably occurred during the polymerization of p-DEB. This was indicated by a decrease in the molecular weights and polydispersity index (${\bar{M}}_{w}/{\bar{M}}_{n}$) of the polymer with an increase in the concentration of the monomer. The termination of the polymer chain occurred due to the transfer of acidic hydrogen from the π-bound p-DEB into the growing chain (Figure 19). In accordance with the scheme, an unbranched trans-poly-p-DEB was formed. However, the article does not investigate the intramolecular structure of the polymer by any spectral methods. Moreover, this conclusion contradicted the results of works [97,98,99], the authors of which reported on the branched structure of poly-p-DEB synthesized on a Ni catalyst. Later [102], this group of authors reported on the branched nature of polymer chains of poly-p-DEB obtained under similar conditions.

The structure of the initial soluble poly-p-DEB doped with FeCl_3_, as well as the mechanism of doping were investigated by a set of spectral methods (FT-IR, Raman, UV, ^1^H, and ^13^C NMR spectra) in [102]. Doping was carried out through a joint solvent CHCl_3_. The electrical conductivity of the initial poly-p-DEB was 5.5·10^−14^ S·cm^−1^ at room temperature. The addition of 22.6–50.1%_wt_. FeCl_3_ to the polymer resulted in an increase in conductivity to 4.0·10^−5^ S·cm^−1^. A further increase in the doping concentration led to a drop in conductivity to 6.5·10^−6^ S·cm^−1^. According to the authors, this is due to a decrease in the concentration of charge carriers, and not to chlorination (results of XPS and FT-IR spectroscopy). The initial polymer was synthesized in 1,4-dioxane-toluene mixed solvent using $\mathrm{Ni}{\left({\mathrm{PPh}}_{3}\right)}_{2}{\left(C\equiv C–C_{6}H_{4}–C\equiv \mathrm{CH}\right)}_{2}$ in accordance with [98]. The authors of this article [102] have declared a branched structure in Figure 10c for poly-p-DEB, referring to an earlier publication [98]. In the ^1^H-NMR spectra of this polymer, the ratio $H_{\mathrm{ol}+\mathrm{ar}}/H_{\mathrm{eth}}=8/1$ was obtained.

Poly-p-DEB was used to create SAW humidity sensors [103]. The PDEB-based SAW sensor was constructed by applying 10 single-molecule layers of poly-p-DEB on a piezoelectric substrate using the LB method using a polymer solution in CHCl_3_. The frequency characteristics of the SAW sensors were studied in the temperature range of 30–90 °C in the relative humidity range of 20–85%. A linear decrease in frequency with increasing temperature is detected; at the same time, the dependence of frequency shift on temperature was −5 kHz/°C. Thus, it became necessary to measure temperature and humidity in order to compensate for the influence of temperature on the sensor response. The frequency of the poly-p-DEB sensor decreased almost linearly with increasing humidity in the 20–85% relative humidity range. The sensitivity at 22 and 30 °C was −0.4 and −0.36 kHz/RH%, respectively (Figure 20). The poly-p-DEB sample was synthesized in accordance with [95] in the presence of $\mathrm{Ni}{\left({\mathrm{PPh}}_{3}\right)}_{2}{\left(C\equiv C–C_{6}H_{4}–C\equiv \mathrm{CH}\right)}_{2}$ under an N_2_ atmosphere for 6 h at 25 °C. That is why it had to have a branched structure in Figure 10c, as was found in an earlier article [95]. However, the authors of this article [103] for some reason indicated the linear structure of the polymer in Figure 3, without confirming this with any additional studies.

A series of air- and moisture-resistant nickelocene acetylides catalysts of the general formula (π-C_5_H_5_)LNi(C≡CR) (where L = PPh_3_, PBu_3_; R = p-C_6_H_4_C≡CH, C_6_H_5_, H) was studied in [104] during polymerization of p-DEB in DMSO or pyridine at temperatures of 40 or 60 °C for a time of 6 or 24 h. As a result, soluble polymers with a high yield were obtained (19–85.2%), with high values of ${\bar{M}}_{w}$ = 10,500–23,400 and ${\bar{M}}_{w}/{\bar{M}}_{n}\;$= 2.7–4.3. Complexes containing PPh_3_ as a ligand were more active than complexes containing PBu_3_. The authors explained this by the greater basicity and the associated σ-donor ability of PBu_3_, which increases the density of the electron cloud of the nickel-carbon bond and reduces the reactivity of this bond. The catalytic activity of the complex increased with an increase in the degree of conjugation of the corresponding alkynyl ligand. According to the results of ^1^H-NMR spectroscopy, the ratio $H_{\mathrm{ol}+\mathrm{ar}}/H_{\mathrm{eth}}=8/1$, characteristic of branched poly-p-DEB, was obtained instead of 5/1 for a linear polymer. According to the authors, the branched structure of poly-p-DEB was additionally confirmed by a higher polydispersity index (${\bar{M}}_{w}/{\bar{M}}_{n}\;$= 2.7–4.3) and smaller Mark–Houwink constants (value a = 0.26). The presence of cis- or trans-isomers has not been discussed.

In [105], the features of the polymerization of p-DEB in pyridine in the presence of $\left(\pi –C_{5}H_{5})\left({\mathrm{PPh}}_{3}\right)\mathrm{Ni}(C\equiv {\mathrm{CC}}_{6}H_{4}C\equiv \mathrm{CH}\right)$ are considered in more detail. The influence of the concentration of monomer and catalyst, gas medium (N_2_, CO, H_2_, air, vacuum), temperature (30–50 °C), additives (H_2_O, EtOH, DMSO, Et_3_N, hydroquinone, CCl_4_, Ph_3_N,), polymerization time (1–18 h) was studied for yield (up to 86%), ${\bar{M}}_{w}$ (up to 55,700) and ${\bar{M}}_{w}/{\bar{M}}_{n}$ (1.42–8.21). It should be noted that the addition of CCl_4_ and Ph_3_N to the reaction medium, as well as polymerization in the CO medium, completely blocked the polymerization process. Carrying out the reaction in pyridine for 6 h at concentrations of [Cat] = 0.01 mol/L and [M] = 1.5 mol/L led to the appearance of an insoluble fraction (at 45 °C) or even to a loss of solubility (at 50 °C). The trans-polymer had branched macromolecules, which was confirmed by the ratio $H_{\mathrm{ol}+\mathrm{ar}}/H_{\mathrm{eth}}=8/1$. The polymer had typical C≡CH group signals in the IR spectra: the ν_C≡CH_ band at 3293 cm^−1^ and the ν_C≡C_ band at 2106 cm^−1^.

The same soluble branched trans-poly-p-DEB synthesized using $\left(\pi –C_{5}H_{5})\left({\mathrm{PPh}}_{3}\right)\mathrm{Ni}(C\equiv {\mathrm{CC}}_{6}H_{4}C\equiv \mathrm{CH}\right)$ according to the method [104,105] was used in [106] in the study of its iodine doping. The doping mechanism of poly-p-DEB has been investigated using UV-vis, FT-IR, FIR, Raman, XPS, and ESR spectroscopies. In the EPR spectra, a signal with a g-factor close to the g value of the free electron was observed. The concentration of unpaired spins slightly increased in the doped poly-p-DEB. However, it was less than expected: not every ligand molecule formed one poly-p-DEB radical cation. The authors believed that the formation of sufficiently stable $I_{5}^{-}$ particles occurred during doping. After doping with iodine, the conductivity of poly-p-DEB increased exponentially depending on the doping concentration by several orders of magnitude and reached a value of 10^−4^ S·cm^−1^ (Figure 21).

The processes of curing soluble homopolymers of p-DEB, m-DEB, and their copolymer, as well as the ongoing structural changes in the polymers, were compared using DSC and FTIR analysis [107]. Poly-p-DEB had a higher curing rate than poly-m-DEB. The authors reported that a “nickel catalyst” was used for polymerization, which, according to the authors, contributed to the formation of a highly branched poly-p-DEB having cis-S-trans polyene fragments and phenylene fragments due to the cyclotrimerization reaction. However, Figure 4 of this article shows only cis-polyene without aromatic fragments (Figure 22).

The fact of the formation of cis-polyenes is surprising and of interest, since the widely used and various Ni-containing catalysts Ni(C_5_H_7_O_2_)_2_ Ph_3_P, $\mathrm{Ni}{\left({\mathrm{PPh}}_{3}\right)}_{2}{\left(C\equiv C–C_{6}H_{4}–C\equiv \mathrm{CH}\right)}_{2}$ and $\mathrm{Ni}{\left({\mathrm{PPh}}_{3}\right)}_{2}(C\equiv C–C_{6}H_{5})$ provide the synthesis of trans-polyenes. The results of any spectral studies of polymers are not given in the article.

In [108], this group of authors studied the rheological properties of arylacetylene prepolymers, probably of the nature that were studied in the previous article [107]. The prepolymers were a Newtonian liquid in the studied temperature range. The viscosity of the prepolymers changed little during 90 min at 60–80 °C, but gradually increased at room temperature.

It was shown in [100] that in the presence of (π-C_5_H_5_)(PPh_3_)NiCl, polymerization of p-DEB proceeded much worse than in the presence of a similar and described above catalyst $\left(\pi –C_{5}H_{5})\left({\mathrm{PPh}}_{3}\right)\mathrm{Ni}(C\equiv {\mathrm{CC}}_{6}H_{4}C\equiv \mathrm{CH}\right)$. Under the same synthesis conditions ([cat] = 0.02 M; [M] = 1.5 M; DMSO; time: 24 h; temperature: 30 °C), the conversion was 6.5 and 42.8%, respectively. The branched π-conjugate polyene structure was indicated by the ratio $H_{\mathrm{ol}+\mathrm{ar}}/H_{\mathrm{eth}}=8/1$ according to FT-IR and ^1^H-NMR spectroscopy.

Thus, the analysis of numerous publications devoted to the synthesis and use of the $\mathrm{Ni}{\left({\mathrm{PPh}}_{3}\right)}_{2}{\left(C\equiv C–C_{6}H_{4}–C\equiv \mathrm{CH}\right)}_{2}$ catalyst leads to the final conclusion that the use of this catalyst leads to the synthesis of branched soluble trans-poly-p-DEB structure in Figure 10c.

The influence of ligands on the activity of palladium catalytic complexes was considered in [99,109]. In the presence of $\mathrm{Pd}{\left({\mathrm{PPh}}_{3}\right)}_{2}{\left(C\equiv C–C_{6}H_{4}–C\equiv \mathrm{CH}\right)}_{2}$ and $\mathrm{Pd}{\left({\mathrm{PPh}}_{3}\right)}_{2}{\left(C\equiv C–{\mathrm{CH}}_{2}\mathrm{OH}\right)}_{2}$ complexes, the conversion of p-DEB was 79.9 and 58.2%, respectively, at 60 °C for 18 h in pyridine. The synthesized poly-p-DEBs had values of ${\bar{M}}_{w}$ = 18,000 and 15,000, respectively, as well as wide ${\bar{M}}_{w}/{\bar{M}}_{n}$ = 3.8 and 3.3. At the same time, the substitution of the PPh_3_ ligand for PBu_3_ in these complexes led to a complete loss of activity of both catalytic complexes (conversion of 0% even in 88 h). The authors associated this with higher basicity and the interconnected donor capacity of PBu_3_, which increases the electron density of the metal–carbon bond and reduces the reactivity of this bond. According to the results of FT-IR, ^1^H, and ^13^C NMR spectroscopy, the polymers had a branched structure, as in the case of similar nickel-based complexes.

Nonlinear optical properties were studied in [110] using soluble poly-p-DEB synthesized using the $\mathrm{Pd}{\left({\mathrm{PPh}}_{3}\right)}_{2}{\left(C\equiv C–C_{6}H_{4}–C\equiv \mathrm{CH}\right)}_{2}$ complex with a conversion of 80%, probably in pyridine. Poly-p-DEB had ${\bar{M}}_{w}\;$= 4670, ${\bar{M}}_{w}/{\bar{M}}_{n}\;$= 3.77 and had a heat resistance of up to 420 °C. The authors indicated the trans-polyene structure of the branched polymer in the figure but did not provide any evidence for this. Apparently, this statement is true, and the structure of the polymer corresponds to the structure of the polymer synthesized as in [109]. The polymer was characterized by a nonlinear optical susceptibility of the third order χ^(3)^=6.7·10^−13^ esu and a second-order hyperpolarizability having γ = 1.8·10^−31^ esu in the non-resonant region.

The effect of phosphine and alkynyl ligands bound to metal atoms on the polymerization of p-DEB was studied in [111] on the example of PdL_2_ (C≡CR) catalytic complexes, where (L = PPh_3_, PBu_3_; R = p-C_6_H_4_C≡CH, C_6_H_5_, H, CH_2_OH, CH_2_OCOCH_3_, CH_2_OCOC_6_H_5_, CH_2_OCOC_6_H_4_OH-o) resistant to air and moisture. During 15–18 h at a temperature of 60 °C, the yield of soluble poly-p-DEB reached 56–84% in the presence of palladium complexes containing the PPh_3_ ligand. Complexes containing the PBu_3_ ligand were inactive (polymer yield 0%), as in the case of the Ni analog [109]. Complexes containing polar alkynyl ligands were more active than corresponding complexes containing nonpolar ligands. The catalytic activity of complexes containing nonpolar alkynyl ligands increased with an increase in the degree of alkynyl ligand conjugation. The polymers had values of ${\bar{M}}_{w}\;$= 1.1–1.8·10^4^ and a wide molecular mass distribution ${\bar{M}}_{w}/{\bar{M}}_{n}\;$= 2.6–3.8. The authors did not discuss the features of the intramolecular structure of poly-p-DEB, but the polymer chains should be branched since the synthesis was carried out in accordance with [109,110].

The catalytic activity of numerous transition metals of the VIII group in the form of chloride and acetylide complexes ML_2_(C≡CC_6_H_5_)_2_ and ML_2_Cl_2_ (M = Co, Ni, Pd, and Pt; L = PPh_3_ and PBu_3_) was compared in [109]. Polymerization of p-DEB was carried out in a medium of various solvents (dioxane/toluene, HNEt_2_/toluene, DMSO, pyridine), at a temperature of 25–60 °C for 3–24 h. Poly-p-DEB was synthesized with a conversion of 51–90%, ${\bar{M}}_{w}\;$= 1.2–2.6·10^4^ and ${\bar{M}}_{w}/{\bar{M}}_{n}\;$= 1.8–3.2. On the whole, the catalytic activity of the complexes increased in the order Pt < Pd < Ni < Co. It was found that the complexes Pt(PPh_3_)_2_Cl_2_, Pd(PBu_3_)_2_Cl_2_, Pt(PBu_3_)_2_Cl_2_, Pt(PPh_3_)_2_(C≡CC_6_H_5_)_2_, Pd(PBu_3_)_2_(C≡CC_6_H_5_)_2_, Pt(PBu_3_)_2_(C≡CC_6_H_5_)_2_ had no catalytic activity. The structure was determined using IR, ^1^H, and ^13^C-NMR spectra. The ratio of olefin and aromatic protons to ethynyl protons was $H_{\mathrm{ol}+\mathrm{ar}}/H_{\mathrm{eth}}=8/1$. This indicated the branched nature of polymer chains. The type of transition metals did not affect the structure of trans-polyene branched polymers with side groups p-C_6_H_5_C≡CH. In the example of the IR spectrum of poly-p-DEB synthesized in the presence of $\mathrm{Co}{\left({\mathrm{PBu}}_{3}\right)}_{2}{\left(C\equiv C–C_{6}H_{5}\right)}_{2}$ given in the article, in addition to the band at 3293 cm^−1^ (ν_C≡CH_), there was a band at 2106 cm^−1^ (ν_C≡C_).

Insoluble cross-linked polycyclotrimers (Figure 23) were obtained with quantitative yield by polycyclotrimerization of p-DEB catalyzed by ${\mathrm{Co}}_{2}{\left(\mathrm{CO}\right)}_{8}$ in anhydrous 1,4-dioxane in an argon atmosphere. The reaction was carried out initially at room temperature with further heating at 125 °C for 1 h [112]. In accordance with the above polymerization scheme (Figure 23), C≡CH bonds disappeared. For this reason, the characteristic vibration band ≡C–H at 3300 cm^−1^ disappeared for poly-p-DEB. The brown polymer showed no signs of decomposition up to 360 °C and had a BET surface of about 1000 m^2^g^−1^, and a pore size of ≈0.83 nm.

Homopolymerization of p-DEB was carried out on a complex catalyst ${\left[{\mathrm{Co}}_{2}{\left(\mathrm{CO}\right)}_{6}\right]}_{2}\cdot \left(\mathrm{PhC}\equiv C–C\equiv C–\mathrm{Ph}\right)$ in benzene, toluene, or N-MP with a conversion of 8–100% [113,114]. At the ratio [M]_0_/[I]_0_ = 62, insoluble, brown, paramagnetic polymers were formed between 5 and 120 min. At short polymerization times τ (up to 40 min in boiling benzene), a finely dispersed gel appeared, which was suspended in the reaction system. With an increase in the reaction time, an insoluble product was already formed in the reaction system. An interesting observation was made: polymerization occurred at a high rate exclusively during the boiling of the reaction mixture. If the reaction was carried out at a temperature below the boiling point of the solvent, then the conversion Y was very small (Table 1). If the reaction mixture was sealed into ampoules and heated to the boiling point of benzene, then boiling did not occur due to temperature depression due to an increase in pressure in the ampoule. At the same time, there was no polymerization. From these results, it followed that only boiling can ensure the reaction on this catalyst, apparently due to any cavitation phenomena.

According to FT-IR, ^1^H and ^13^C-NMR spectra, poly-p-DEB was a polyene with lateral ethynylphenyl substituents, some of which reacted to form a 3D homopolymer grid.

Extremely scarce and contradictory information is given in [115] on the synthesis of insoluble poly-p-DEB in the presence of Nb catalysts. Probably, the polymerization conditions were the same as for the polymerization of α, ω-alkadiynes studied in this article: NbCl_5_ catalyst; [M]_0_/[I]_0_ = 50/1; solvents benzene (toluene); 55–65 °C; N_2_ atmosphere; polymerization time was 5 h.

Very limited information on the polymerization of p-DEB in the presence of $\mathrm{CpCo}{\left(\mathrm{CO}\right)}_{2}$ complexes under the action of UV radiation or TaCl_5_/Ph_4_Sn is available in [21]. The authors reported that polymerization was fast. The resulting polymer products were completely insoluble due to the presence of crosslinking reactions, which indicated the appearance of branching directly during the polymerization process.

The polymerization of p-DEB in the presence of TaCl_5_/Ph_4_Sn is considered in more detail in [116]. The reaction was carried out at an initial ratio of [M]_0_ = 0.6 mol L^−1^, [TaCl_5_]_0_ = 0.015 mol L^−1^, [Ph_4_Sn]_0_ = 0.015 mol L^−1^ in benzene at room temperature for 24 h. The conversion rate was 97%. Solid yellow-orange polycyclotrimers did not dissolve in THF, CH_2_Cl_2_, CHCl_3,_ and benzene and had a crosslinked structure. The ^13^C-CP/MAS NMR and IR spectra demonstrated an insignificant number of unreacted groups in the polymer –C≡CH. Signals in the area of 70–85 ppm and bands at 3290 cm^−1^ (ν_C≡CH_) and 2100 cm^−1^ (ν_C≡C_) were recorded, respectively. The authors believed that the crosslinking of polymer chains occurred due to the formation of 1,3,5- and 1,2,4-trisubstituted aromatic linkers (Figure 24) since the formation of such trimers occurs during cyclotrimerization of acetylene derivatives in the presence of TaCl_5_ [117,118].

This version was confirmed by the authors [116] by the pattern of substitution of aromatic rings from IR in the region of 600–900 cm^−1^, as well as in the region of 1660–2000 cm^−1^ (overtones and bands of combined tones). N_2_ isotherms demonstrated that adsorption/desorption hysteresis occurs already at low equilibrium pressures. The CO_2_ isotherms demonstrated that the time allotted for measurement affects both the maximum adsorption capacity and the hysteresis during desorption. For poly-p-DEB, the values of the micropore volume V_MI_ = 0.472 cm^3^ g^−1^, S_BET_ = 1299 m^2^ g^−1^, the maximum sorption capacity (the highest amounts of nitrogen and hydrogen adsorbed), for H_2_ and CO_2_, respectively, a(H_2_)= 1.26%_wt_. (100 kPa, 77 °K) and a(CO_2_)= 10.8%_wt._ (100 kPa, 273 °K).

In a very short publication [119], it was reported about a vigorous exothermic polymerization of p-DEB in the presence of PR_3_ (R = Me_2_NO, Et, Ph, Bu,) with the formation of a black insoluble substance with a conversion of up to 90%. The authors noted that the conversion was always high “at any Monomer/Phosphine ratios, even with catalytic amounts of the phosphorus component”. Of the experimental conditions, the authors cited only time (5–30 min) and temperature (90 °C or 130 °C in one experiment). In the presence of dimethylbenzylamine, the conversion was only 30% during 60 min at 130 °C. The authors suggested the zwitterionic nature of the initiation of the polymerization process, as it occurs in the presence of amines and phosphines [120].

The preparation of poly-p-DEB by electroinitiated anionic polymerization in an inert atmosphere at 10 °C in polar solvents was described in [121]. The current intensity was changed in the range of 0–70 mA. The efficiency of electrolytes decreased in the series (C_4_H_9_)_4_ Cl > (CH_3_)_4_NClO_4_ > (C_2_H_5_)_4_NClO_4_. The soluble fraction conversion was up to 30%. It was found that polymerization proceeded with good results in N,N-dimethylformamide and especially in N,N-dimethylacetamide. Separated oligomers were obtained in methylene chloride. In acetonitrile, the electrolyte decomposed at any current; the characteristic color of the polymer did not appear, which indicated the absence of polymerization. According to the authors, electropolymerization began with a direct reduction of the monomer on the electrode surface. Then the reaction proceeded according to the classical anionic mechanism:$$M+e^{-}\to M^{-}\;2M^{-}\to^{-}M-M^{-}$$

The authors pointed out that only one ethynyl group of the monomer participates in the reaction, which, according to the authors, “indicates the presence of a large number of unreacted triple bonds in accordance with absorption at 3250–3260 cm^−1^ in the IR spectra”. However, they did not provide any direct evidence for the synthesis of a linear polymer, including an assessment of the results using the ^1^H-NMR spectra they obtained. Nevertheless, it is most probable that branched poly-p-DEB with a reduced number of –C≡CH groups was obtained in this work, since at the end of the article, it is indicated that “the second ethynyl group of the monomer molecule is transformed to a lesser extent, obeying the same principle of reduction on the electrode surface”. This is confirmed by the fact that an insoluble fraction is present in the synthesized polymers, which are sequentially formed from branched polymer molecules.

In an early work [122], p-DEB was polymerized in CH_3_NH_2_ in ampoules ([DEB]/[CH_3_NH_2_] = 1/5 mol) at 260 °C. In 1 or 5 h, an insoluble polymer was obtained with a conversion rate of 87 and 94%, respectively. There are no spectral studies of polymers in the article. Nevertheless, according to the authors, poly-p-DEB polyene chains have two types of side substituents (including phenylene fragments) in accordance with the polymerization scheme (Figure 25).

In our opinion, in this case, most likely, zwitter-ionic polymerization took place with the initial formation of a linear polyene polymer. In this case, CH_3_NH_2_ amine played the role of not only a solvent but also a classical zwitter-ion initiator. Subsequently, due to the extremely high temperature, a regular crosslinking of the chains occurred due to the opening of triple bonds in the –C≡CH groups. A similar process of pyrolysis of DEB polymers was later described in numerous publications. As a result, the proposed structure of an insoluble polymer having phenylene fragments seems implausible. Probably, such a structure (Figure 25) is formally borrowed from the publications of these authors, who at that time studied catalytic cyclotrimerizing complexes ${\left[{\left(\mathrm{RO}\right)}_{3}P\right]}_{3–4}\cdot \mathrm{CoX}$ (where R = Alk_C≤6_, Hal = Cl, Br, I) in articles [67,74]. Most likely, the polymer structure can be represented by the formula shown in Figure 15.

At [123] carried out soft chemical functionalization of the silicon surface using p-DEB in the dark at room temperature. In this case, only individual p-DEB molecules are fixed on the silicon surface without the formation of polymers or even oligomers. The subsequent click reaction with azidomethylferrocene made it possible to create a ferrocenyl-modified coating.

### 2.5. Synthesis of Linear Unbranched Poly-DEB

Thus, as a rule, only branched and/or insoluble poly-p-DEB was synthesized by various groups of researchers by direct polymerization of p-DEB. It seemed that direct chemical (including electrochemical) synthesis of linear, unbranched poly-p-DEB by selective polymerization of the p-DEB monomer at only one C≡C bond was impossible. For this reason, it may not be possible to synthesize clusters along a one-dimensional poly-conjugated chain.

However, in 1981, we reported the synthesis by our research group of a linear soluble polymer p-DEB in a solution of hexamethylphosphoramide (HMPT) in the presence of an anionic initiator [124,125]. Later, in [33,126], the features of the intramolecular structure of this polymer were examined in more detail using IR, ^1^H- and ^13^C-NMR spectroscopy, and HPLC methods. The effect of p-DEB polymerization conditions on the yield and properties of poly-p-DEB is shown in Table 2 [33].

It was found that the polymerization of p-DEB in HMPT produces a completely soluble linear unbranched polymer. The ratio $H_{\mathrm{ol}+\mathrm{ar}}/H_{\mathrm{eth}}=5/1$ was found in its ^1^H-NMR spectrum. This corresponded to the presence of one ethynyl group in each elementary link of the macromolecule. The increase in polymerization time did not lead to a significant increase in polymer yield. However, at the same time, the number of C=C bonds in the polymer decreased (an increase in the ratio of $H_{\mathrm{ol}+\mathrm{ar}}/H_{\mathrm{eth}}=5.3/1$). Decreasing the solvent polarity resulted in only weakly branched polymers. For polymers with a deficiency of –C≡CH groups, chain transfer to the monomer occurred during polymerization (Figure 26), which explained this deficiency.

In [31,32,33], the principal possibility of modifying synthesized poly-p-DEB was demonstrated by the interaction of groups –C≡CH with reagents containing heteroatoms: ${\mathrm{Co}}_{2}{\left(\mathrm{CO}\right)}_{8}$, B_10_H_14_, CuCl. As a result, poly-p-DEB derivatives having copper, boron, and cobalt in some links were synthesized and analyzed (Figure 1). This approach demonstrated the fundamental possibility of creating various clusters along the main polyene polymer chain. In addition, synthesized anionic poly-p-DEB has been used to create and modify various materials.

Poly-p-DEB, as well as other phenyl-containing polyacetylenes, was used in the creation of colored, dense, homogeneous, photo-resistive skin layers with a thickness of 0.1–10 microns by vacuum spraying these polymers. The resulting layers had both negative and positive photo-resistive properties, depending on the method of their processing [127,128]. It was found that the short-term irradiation of the layers with UV light in a vacuum through a photomask led to the appearance of a latent image due to the intermolecular crosslinking of polymers in the irradiated areas. After the layers were heated in a vacuum at an optimal temperature of 200–350 °C, the non-irradiated uncrosslinked part of the polymers evaporated. As a result, a visible contrast image appeared. On the contrary, the long-term irradiation of the sprayed films with UV light through a photomask in the air led to the disappearance of the areas of the sprayed layers in the places that the light hit and to the appearance of a contrasting positive image. This process was explained by the authors [127,128] by the formation of ozone from air oxygen under the action of UV rays and the subsequent course of the oxidation reaction of unsaturated bonds of sprayed polymers by ozone. In this case, in accordance with the values of the rate constants of the interaction of ozone with organic compounds [129] (Table 3), an ozone attack on the C=C bonds of polyene chains is most probable. Only this reaction of the destruction of polyene chromophores forming the sprayed layers will lead to the appearance of a visible image.

In [33,130,131,132], a significant increase in the thermal oxidation stability of meshes made of three industrial oligoesteracrylates with poly-p-DEB additives was demonstrated. At the same time, the reduction in mass loss of crosslinked compositions with poly-p-DEB was most significant at the highest temperatures. For example, the crosslinked TGM–3 (CH_2_=C(CH_3_)–C(O)–(OCH_2_CH_2_)_3_–O–C(O)–C(CH_3_)=CH_2_) had 75% mass loss at 380 °C. The addition of 20% poly-p-DEB to the uncured resin made it possible to increase the maximum temperature of loss of 75% of the mass at 560 °C. On the contrary, the addition of polyphenylacetylene (a structural analog of poly-p-DEB, which does not have groups –C≡CH) had no effect on the heat resistance of the crosslinked oligoesteracrylate. Such compositions have also been investigated as anaerobic sealants in nut-bolt threaded connections. After curing the compositions, the average torque was measured while unscrewing the nut in one turn. For cured TGM-3 without additives, this value for operating temperatures of 20, 200, and 300 °C was 3, 0.12, and 0 kg·m^−1^, respectively. The addition of 20% poly-p-DEB to the initial polyester increased the torque of the cured compositions to the values of 3, 0.87, and 0.12 kg·m^−1^.

A significant increase in the oxygen resistance of industrial-oriented carbon fibers modified with poly-p-DEB was found [33,133,134]. To do this, the fibers were impregnated with a solution of a composition of poly-p-DEB (5%) and a carborane-containing polymer (5%) in THF. The samples were dried in air and pyrolyzed in N_2_ medium at 500–2400 °C. The smallest weight loss and decrease in oxygen uptake were observed for samples treated at 1500 °C. The authors explained this effect by the formation of BN on the carbon fiber surface as a result of the interaction of the nitrogen medium with the carborane core decaying during heating, as well as the additional effect of migration into the fibers of boron, which is an effective catalyst for graphitization.

In a patent published later [135], it was proposed to use carbenes (Figure 27a) of various metals (Mo, W) for polymerization of p-DEB in a nitrogen atmosphere, wherein M is tungsten and molybdenum, R_5_ is a C_1–10_ alkoxy group or a C_6–10_ aryl group, and R_6_ is a C_1–10_ alkyl group, a C_3–10_ cycloalkyl group, a C_6–10_ aryl group, or a hydrogen atom. At the same time, it is proposed to use these catalysts for the polymerization of an almost unlimited number of diethynylbenzenes of the general structure in Figure 27b, where R_1_, R_2_, R_3,_ and R_4_, each independently, is a hydrogen atom, C_1–10_ alkyl group, C_6–10_ cycloalkyl group, aryl group or halogen atom, and the ethynyl group is either in the meta-position or para-position to another ethynyl group. As a result, according to the authors, ref. [135] only a linear polymer of the structure in Figure 27c should be obtained, although the section “SUMMARY OF THE INVENTION” reports the occurrence of a gel with a conversion of more than 50%.

The synthesis of the DEB homopolymer using tungsten and molybdenum in specific catalytic systems was described only in four patent examples. The authors produced a polymer with a conversion rate of 4–6% and ${\bar{M}}_{w}$ = 3400–25,000. Only in the case of the use of methoxyphenylcarbenpentacarbonyl tungsten, the conversion reached 40%. An integral ^1^H-NMR spectrum was given for this polymer, on the basis of which the authors believe that the polymer has a linear structure as in Figure 27c. However, our analysis of the given ^1^H-NMR spectrum showed that there is a ratio of $H_{\mathrm{ol}+\mathrm{ar}}/H_{\mathrm{eth}}=4.6/1$ instead of 5/1 for a linear structure. The lack of aromatic protons indicated that the spectrum probably belonged to a polymer synthesized from a diethynylbenzene derivative containing a substituent (or substitutes) in the benzene ring instead of hydrogen. Interestingly, according to the authors, this spectrum also characterized the structure of other p-DEB polymers, regardless of the use of other types of catalysts. Moreover, this spectrum was attributed to poly-p-DEB synthesized in the presence of n-Bu_4_Ti/Et_3_Al (1/4 mol) with a yield of 5% and having ${\bar{M}}_{w}\;$= 20,000. The polymerization of p-DEB in the presence of WCl_6_ and n-Bu_4_Ti/Et_3_Al was characterized by a very small conversion and was accompanied by the formation of an insoluble gel. These features indicate a high probability of branching in the polymer already at the beginning of polymerization, followed by the formation of a polymer mesh. Indeed, the formation of insoluble p-DEB polymers in the presence of Ziegler-Natta catalysts was observed much earlier [66,68,69]. Thus, the sum of the above facts makes us question the proposed linear structure of the polymers of p-DEB synthesized in [135]. Poly-p-DEB polymers synthesized in [135] have been proposed to be used in composite resins in combination with oligomers that contain two or more –C≡CH terminal groups [136].

Much later, complexes based on cobalt salts were used in [137]. In the presence of a freshly prepared complex of ${\mathrm{CoHal}}_{2}\cdot 2{\mathrm{PBu}}_{3}$ (Hal = Cl, Br; [M]/[Cat] = 100, mol) at 25 °C, in Et_2_NH medium, during 10 h in an N_2_ atmosphere poly-p-DEB were obtained with yields up to 73.1 and 66.5%, values ${\bar{M}}_{w}\;$= 5100 and 5800, ${\bar{M}}_{n}$ = 2200 and 3900, $\frac{{\bar{M}}_{w}}{{\bar{M}}_{n}}$ = 2.32 and 1.49, respectively, for the Cl- and Br-derivatives. Solubility in THF was 80%. This indicated the emergence of 3D meshes formed from intermediate-branched polymers during polymerization. In the presence of ${\mathrm{CoHal}}_{2}\cdot 2{\mathrm{PPh}}_{3}$, only traces of the polymer were observed. Polymerization did not take place in the presence of ${\mathrm{CoCl}}_{2}\cdot 2H_{2}O$. It was shown that with increasing polymerization time, the yield of poly-p-DEB (the catalyst $\;{\mathrm{CoCl}}_{2}\cdot 2{\mathrm{PBu}}_{3}$), its values ${\bar{M}}_{w}$ and ${\bar{M}}_{w}/{\bar{M}}_{n}$ increased, while the solubility of polymers in THF decreased. The latter indicated the formation of a grid of intermediate-branched polymers formed during polymerization. Indeed, in the ^1^H-NMR spectra of polymers with ${\bar{M}}_{w}\;$= 10,000, 13,000, and 18,000 (${\mathrm{CoCl}}_{2}\cdot 2{\mathrm{PBu}}_{3}$; 2, 10, 15 h, respectively) the ratio of the signal intensities of $H_{\mathrm{ol}+\mathrm{ar}}/H_{\mathrm{eth}}$ was 5/1, 5.1/1 and 5.6/1, respectively. At the beginning of polymerization, poly-p-DEB had a linear structure, turning into a weakly branched structure at a time of up to 10 h. A highly branched structure appeared at a time of more than 10 h. At the same time, the solubility of the polymer deteriorated naturally due to an increase in the number of cross-links (Table 3). That is, the initial linear structure gradually became branched with polymerization time, similar to how it was previously observed in [33,124,125]. The results of ^1^H-NMR spectroscopy presented by the authors [137] convincingly prove the possibility of synthesizing linear non-branched poly-p-DEB in the presence of ${\mathrm{CoCl}}_{2}\cdot 2{\mathrm{PBu}}_{3}$ with a short synthesis time. Although, at the same time, there is a need to explain the reason for the existence of a large value of ${\bar{M}}_{w}/{\bar{M}}_{n}$ in a sample that was synthesized for 2 h.

For this polymer, the third-order nonlinear optical properties of the samples in THF and toluene were measured using the method of direct degenerate four-wave mixing with or without a pump beam was studied in [138]. The pump beam was used to excite molecules to excited states. The second-order molecular hyperpolarizability of the ground state γ_g_ = 10^−31^ esu and the excited state γ_e_ = 10^−29^ esu. The authors believed that they were working with a linear poly-p-DEB, which has one free ethynyl group in each link. Unfortunately, they did not prove the intramolecular structure of the poly-p-DEB they proposed. They did not even indicate the duration of polymerization, which should significantly affect the structure of poly-p-DEB in accordance with [137].

Using tetrahydrofuran solutions of poly-p-DEB as an example, the effects of phase modulation during self-diffraction in nonlinear optical media were studied in [139]. Some interesting results predicted by the theory have been observed. For example, the scattering intensity of a sample with a lower concentration may be greater than that of a sample with a higher concentration at a suitably high pumping intensity. The article does not specify a catalyst. The structure of the polymer under study is not given, and there are no references to the literary source according to which the polymer under study was synthesized. The first two co-authors in this paper were also the first co-authors of an article published later in 1998 [138] in which the ${\mathrm{CoCl}}_{2}\cdot 2{\mathrm{PBu}}_{3}$ complex was used as a catalyst. Probably, in the article [139], a polymer synthesized using this complex was used.

### 2.6. Intramolecular Structure of Substituted Polyacetylenes

Thus, in poly-p-DEB molecules, there are reactive substitutes –PhC≡CH, which can make it possible to modify the synthesized polymer with various heteroatoms. However, the possibility of modification reactions depends on the steric accessibility of C≡C or ≡C–H bonds, and this, in turn, depends on the isomerism of any substituted polyacetylene, including poly-p-DEB.

The isomeric composition of polyacetylenes is influenced by the type of polymerization catalyst used. For example, the microstructure of a polymer obtained from HC≡C–C_6_H_4_–C≡C–Si–i–Pr_3_ in the presence of ${\left[\mathrm{Rh}\left(\mathrm{cod}\right)({\mathrm{OCH}}_{3})\right]}_{2}$ was attributed by the authors [140] to the cis-transoid type (NMR: 94% cis) with “head-to-tail” attachments. When MoCl_5_ was used, the content of cis-structures decreased to 70% [141]. Thermal polymerization in the 4-methoxyphenyl-ethynyl ketone mass resulted mainly in a polymer with a rare type of attachment “head to head-tail to tail” (HH-TT). The traditional HT connection was in smaller quantity [142]. In the case of electrically initiated polymerization [121], the authors only pointed out the existence of both cis- and trans-conformations of double bonds of the main polymer chain. A high content of cis-isomers was observed [143] in fluorophenylacetylene polymers synthesized in the presence of ${\left[\mathrm{Rh}\left(\mathrm{cod}\right)({\mathrm{OCH}}_{3})\right]}_{2}$. On the contrary, the use of ${\mathrm{WOCl}}_{4}\cdot 2{\mathrm{Ph}}_{4}\mathrm{Sn}$ resulted in a mixture of cis-trans isomers. The use of monophosphine-ligated Pd complexes [144] made it possible to obtain cis polymers of disubstituted acetylenes.

The need to evaluate the intramolecular structure of substituted polyenes consists of the correct interpretation of the results of studies of the properties of specific synthesized poly-p-DEB polymers. For example, the authors [141] pointed out the discrepancy between the content of cis-units and π-conjugation determined by NMR and UV spectroscopy in a polymer obtained from HC≡CC_6_H_4_C≡C–Si–(i–Pr_3_). They explained this by the presence of conjugation defects in macromolecules. One of the reasons could be the presence of cis-transoid units attached according to the HH-TT principle. The second reason could be the presence of saturation points in polymer chains, probably in the form of groups –CH_2_–. However, using the example of polyphenylacetylene, the authors considered only a limited number of possible structures (Figure 28), which in reality should be more.

Thus, the type of catalyst (initiator), as well as the type of monomer, affect the intramolecular structure of substituted polyacetylenes. In turn, for synthesized poly-p-DEB, it is necessary to have well-defined information about the conformation in order to interpret the research results and to ensure the possibility of creating clusters along the polymer chain. In this case, the properties of the cluster can be controlled by the quantities, size, and nature of the heteroelement (modifier), introduced into the polymer. In the presence of real steric accessibility of phenyl and ethynyl fragments, it is possible in principle to synthesize various metal-containing polymers: macromolecular acetylenides, carboranes, and ethynyl- and arencarbonyl π-complexes. Such modified polymer systems should be expected to have qualitatively new properties. However, any kind of disturbances in the polymer chain will lead to the disappearance of the possibility of a synthesis of such clusters. In addition, knowledge of the steric features of poly-p-DEB is also necessary to clarify the possibility of conducting reactions of 1,3-dipolar cycloaddition or Sonogashira-Hagihara, as mentioned above. Although it may not be necessary to carry out a modification reaction in each unit of the polymer when creating some materials.

### 2.7. Features of the Intramolecular Structure of Poly-p-DEB

In this section, all theoretically possible types of conformations and configurations of poly-p-DEB (cis-trans isomers, head-to-tail attachments, or head-to-head attachments) are considered. Linear poly-p-DEB molecules are rigid rods, for which the concept of segmental mobility is not applicable. Therefore, there are severe restrictions on the steric availability of phenyl and ethynyl fragments in the central units of macromolecules. At the same time, the following groups and bonds are always accessible sterically in the terminal units of the polymer: –Ph–, –C≡C–, ≡C–H. Selective disclosure of only one C=C bond in the DEB should lead to the realization of the polyene structure of the main chain. Theoretically, it is possible to form four types of cis- and trans-conformers with respect to the C=C bond, as well as cis- or trans-isomers with respect to the C-C bond. In addition, different types of attachment of HT or HH-TT units are possible for each conformer (Figure 29).

In order to clarify the steric features of poly-p-DEB, we collected and studied its Stuart-Briegleb molecular models. To obtain reliable results, the polymer chain was composed of 10–12 monomeric links. Analysis of the collected Stuart-Briegleb models [32,33] demonstrated the results:formation of cis-S-transoid in structure 8 is not possible;the trans-S-cisoid in structure 10b cannot be realized when connecting the head-head-tail-tail links;other types of in structures 7, 9, and 10a may be formed.

## 3. Synthesis, Structural Features, and Properties of o-Diethynylbenzene Polymers

Polymerization of o-DEB in the presence of complexes ${\mathrm{AlEt}}_{n}{\mathrm{Cl}}_{3–n}/{\mathrm{TiCl}}_{4}$, ${\mathrm{AlEt}}_{n}{\mathrm{Cl}}_{3–n}/\mathrm{Ti}{\left(\mathrm{acac}\right)}_{3}$ (where n = 1, 2, 3), $\mathrm{Al}{\left(i–\mathrm{Bu}\right)}_{3}/{\mathrm{TiCl}}_{4}$, $\mathrm{Al}{\left(i–\mathrm{Bu}\right)}_{3}/\mathrm{Ti}{\left(\mathrm{acac}\right)}_{3}$ at 0 or 30 °C d for 1 or 4 h was studied in [145]. As a result, benzene-soluble brown heat-resistant polymers with a conversion rate of 8–93% and with a molecular weight ${\bar{M}}_{w}$ = 2090–3500 were synthesized. In the case of ${\mathrm{AlEt}}_{3}/{\mathrm{TiCl}}_{4}$ and ${\mathrm{AlEt}}_{2}\mathrm{Cl}/{\mathrm{TiCl}}_{4}$ complexes, an insoluble fraction (8.3–56%) appeared simultaneously. The polymer synthesized in the presence of $\mathrm{Al}{\left(i–\mathrm{Bu}\right)}_{3}/{\mathrm{TiCl}}_{4}$ was completely insoluble. Using ^1^H-NMR and IR spectra, it was found that there are 32–48% residual ethynyl groups in soluble polymers. Acetylacetonate complexes $\mathrm{Co}{\left(\mathrm{acac}\right)}_{3}$, $\mathrm{Fe}{\left(\mathrm{acac}\right)}_{3}$, $\mathrm{Ni}{\left(\mathrm{acac}\right)}_{2}$, $\mathrm{Cr}{\left(\mathrm{acac}\right)}_{3}$ with AlEt_3_ proved to be ineffective as catalysts. A small catalytic activity was demonstrated by AlEt_3_-VO(acac)_2_. Cationic polymerization in the presence of TiCl_4_ made it possible to synthesize a polymer with a molecular weight ${\bar{M}}_{w}\;$= 2500 or 2900 in a time of 3–5 h; the conversion reached only 6.4%. Cationic initiators BF_3_OEt_2_, AlCl_3_, and AlEtCl_2_ were less effective. Polymerization was more effective in the presence of AlEt_2_Cl (19% in 10 min). The use of a radical initiator of AIBN made it possible to synthesize poly-o-DEB with a conversion rate of only 2.4% in 7 h. In this polymer, 70% of the second ethynyl group remained unreacted. The authors [145] explained the appearance of solubility in hexane of polymers obtained with cationic and radical initiators by different structures of these polymers. Taking into account the results of spectral studies and the construction of molecular models, several types of fragments of poly-o-DEB molecular chains were proposed (Figure 30).

The thermal curing reaction of o-DEB was studied in [62]. The monomer was considered for comparison with other ethynylbenzenes (see the section *liquid-phase polymerization of p-DEB without the use of catalysts*).

In early works [146,147], the thermal polymerization of o-DEB and two of its derivatives obtained by substituting acidic protons ≡CH for the Ph group (R=H, R’=Ph, and R=R’=Ph) was studied. The o-DEB, when heated in a sealed test tube at 140 °C, formed an insoluble brown polymer over a period of 24 h. The authors believed that poly(1,4-naphthylene) was synthesized in accordance with the scheme in Figure 31 (R=R’=H).

However, the use of various physicochemical methods to study the intramolecular structure of the thermal polymer o-DEB and low molecular models (IR spectroscopy, MALDI-TOF MS, solid-state NMR spectroscopy, UV-vis reflectance spectroscopy, and pyrolysis GC-MS data) allowed the authors [148] to review the results published earlier in [146,147]. In accordance with the results obtained, the authors believed that the polymer could not have a poly(1,4-naphthylene) structure. It has a more complex structure implemented in accordance with the proposed Bergman cyclopolymerization scheme (Figure 32).

The metathesis catalyst systems (${\mathrm{MoCl}}_{5}$, ${\mathrm{WCl}}_{6}$, ${\mathrm{MoCl}}_{5}/{\left(n–\mathrm{Bu}\right)}_{4}\mathrm{Sn}$, WCl_6_/(n−Bu)_4_Sn, ${\mathrm{WCl}}_{6}/{\mathrm{EtAlCl}}_{2}$, ${\mathrm{MoCl}}_{5}/{\mathrm{EtAlCl}}_{2}$) proved to be very effective in the polymerization of o-DEB [149]. At [M_0_] = 0.5 or 1.0 mol/L, [M_0_]/[Cat] = 25 or 50 mol, at a temperature of 60 °C during 24 h, the yield of dark red, insoluble in methanol polymers was 82–97%. Using the ${\mathrm{MoCl}}_{5}$ catalyst as an example, under the same experimental conditions, the authors studied the effect of the solvent on the polymer yield. Interestingly, there was a high polymer yield in chlorobenzene, chloroform, and toluene, but only traces of polymers were found in THF, 1,4-dioxane, and DMF. The polymers had poor solubility in chloroform and THF. Apparently, this can be explained by the presence of insoluble fractions in polymer samples. The authors reported that the I_2_-doped polymers had a conductivity at room temperature of about 10^−6^ S·cm^−1^. In their opinion, the polymers had a structure with indenylene fragments (Figure 33).

When using a catalyst ${\left[\mathrm{Rh}\left(\mathrm{nbd}\right)\mathrm{Cl}\right]}_{2}/{\mathrm{Et}}_{3}N$ (component ratio ${\left[\mathrm{Rh}\left(\mathrm{nbd}\right)\mathrm{Cl}\right]}_{2}/{\mathrm{Et}}_{3}N$ = 1/10 or 1/4, [M_0_] = 0.5 mol/L, [M_0_]/[Cat] = 100 or 50 mol, time 24 h, atmosphere Ar, solvents THF or toluene) at a temperature of 30–80 °C, yellowish-brown insoluble poly-o-DEBs were synthesized and had an 18–73% yield [17]. These polymers had indene–type fragments and contained 50–73% of unreacted –C≡C–H groups. On the other hand, the use of ${\mathrm{TaCl}}_{5}/{\mathrm{Ph}}_{4}\mathrm{Sn}$ (component ratio ${\mathrm{TaCl}}_{5}/{\mathrm{Ph}}_{4}\mathrm{Sn}$ = 1/2, [M_0_] = 0.5 mol/L, [M_0_]/[Cat] = 25 or 50 mol, 24 h, Ar, toluene, 80 °C) allowed to obtain an insoluble poly-o-DEB with a different structure with a yield of 79–91%. It was a highly crosslinked mesh of tri–substituted benzene that did not contain the –C≡C–H group. The use of various methods for studying (^1^H and ^13^C-NMR, IR-, UV–vis spectroscopy, GPC, TGA, pyrolysis GC-MS) allowed the authors [17] to propose two different structures of synthesized polymers, respectively (Figure 34).

The review [150] describes the cyclopolymerization processes of various bis-o-diynylarenes, including some of the above-discussed articles on o-DEB. Examples of the use of bis-o-diynylarenes polymers are given. Thus, o-DEB polymers had a complex intramolecular structure, including indenylene fragments, and were insoluble in the overwhelming majority of cases.

## 4. Synthesis, Structural Features and Properties of m-Diethynylbenzene Polymers

As early as 1988, it was discovered that the distillation of m-DEB at high temperatures provides an exothermic reaction and leads to an explosion [151].

Apparently, in [152] there is the first report on the targeted homopolymerization of m-DEB. As a result, an insoluble polymer was obtained, but the authors did not specify a catalyst and did not provide any polymerization conditions. A probable catalyst could be a catalyst from the group ${\left[{\left(\mathrm{RO}\right)}_{3}P\right]}_{n}\cdot \mathrm{CoHal}$, (where R = Alk_C≤6_, Hal = Cl, Br, I), since this group of authors had previously synthesized other polymers using such catalysts [66,67,74,75].

In [12], it was proposed to use m-DEB, as well as other mono- and disubstituted acetylene-substituted aromatic compounds (1-ethynylpyrene, 1-ethynylnaphthalene, 3-ethynylphenanthrene, 4,4′-diethynyldiphenylmethane, and 4,4′-diethynylbiphenyl), for the synthesis of matrix resins necessary for the manufacture of carbon/carbon composites. After curing and pyrolysis, the char yield reached values up to 95%.

The authors of the patent [135] proposed polymerizing m-DEB using as a catalyst a substance obtained by the interaction of WCl_6_ with phenylacetylene in a nitrogen medium in a toluene solution for 30 min at room temperature. The subsequent polymerization reaction of m-DEB (nitrogen, toluene, 30 °C, 3 h) gave a polymer with ${\bar{M}}_{w}$ = 23,000 according to GPC data. The linear structure of polymer chains followed from the ^1^H-NMR spectrum ($H_{\mathrm{ol}+\mathrm{ar}}/H_{\mathrm{eth}}$ = 5/1). However, this fact is doubtful, since, according to the authors [135], p-DEB polymers synthesized using five different catalysts have the same spectrum. In addition, polymerization of p-DEB in the presence of WCl_6_ gave the gel after 30 min. This indicated that the appearance of branching in the still soluble polymer chains was already at the early stages of the process due to the reaction of free groups –C≡C–H. In turn, this should have led to an increase in the $H_{\mathrm{ol}+\mathrm{ar}}/H_{\mathrm{eth}}$ value.

The $\mathrm{Ni}{\left(\mathrm{acac}\right)}_{2}\cdot 2{\mathrm{Ph}}_{3}P$ catalyst was used in [153] for the polymerization of m-DEB in methyl isobutyl ketone at 85 °C. The polymer had ${\bar{M}}_{n}\;$= 2490 and contained 10.1%_wt._ of terminal acetylene groups. The intramolecular structure was not discussed.

Later articles [154,155,156] discussed the problems of using polyarylacetylene (PAA) resin in various carbon-carbon composites. The authors illustrated their considerations with the example of a PAA resin synthesized by the polycyclotrimerization of m-DEB in accordance with the scheme (see Figure 35). The authors report that the Ni content in the cured resin was less than 0.1%_wt_. A specific nickel catalyst was not shown; however, the authors probably had in mind the frequently used $\mathrm{Ni}{\left(\mathrm{acac}\right)}_{2}\cdot 2{\mathrm{Ph}}_{3}P$ as a catalytic system.

It should be noted that for the production of PAA resin, various authors have proposed using various combinations of isomers of diethynylbenzene, phenylacetylene, and other mono- and disubstituted aromatic substances [23,157,158,159,160,161].

The process of thermal curing of resins with arylacetylene terminal groups was studied in [62]. In this article, the initial monomer m-DEB was considered as a comparison with other ethynyl-containing benzenes (see the section *liquid-phase polymerization of p-DEB without the use of catalysts*).

The article [162] was devoted to the study of the features of the polymerization of m-DEB from the point of view of optimal preparation of composite materials based on m-DEB. The article noted that the thermal and catalytic polymerization of m-DEB (temperature, time, and catalyst are not shown) took place with a large heat release and led to the formation of insoluble homopolymers. The authors recommended the preliminary preparation of a prepolymer, the subsequent curing of which occurs with less heat release and the formation of an insoluble black polymer. The enthalpy of polymerization of m-DEB into a crosslinked polymer decreased from ∆H = 2013 J/g to 452.6 J/g when using a prepolymer. The structure was determined using IR spectroscopy (tablets in KBr). It is believed that part of the ethynyl groups forms substituted benzene; the other part gives a grid. As a result of the reaction, phenylene, and cis-polyene fragments are formed in the polymer structure; no trans-structure is formed.

The results of a comparative study of the thermal curing process of soluble homopolymers m-DEB, p-DEB, and their copolymer by DSC and FTIR analysis are presented by this group of authors in [107,108]. It was found that the curing activation energy is higher in poly-m-DEB than in poly-p-DEB. The polymers were synthesized in methyl ethyl ketone using a “nickel catalyst” (not specifically noted), which, according to the authors, provides trimerization. However, the benzene fragments were not depicted in the figure given in the article (Figure 22). It was reported that the absorption peaks of acetylene groups decreased significantly during curing (200 °C, 2 h) and disappeared at a higher temperature (300 °C, 2 h). At the same time, peaks appeared, indicating the appearance of cis- structures.

In [21,163], the synthesis of the insoluble polymer m-DEB was reported, but no specific catalyst and polymerization conditions were specified. Probably, the reaction was carried out in toluene in an atmosphere of dry nitrogen at room temperature or at 60 °C using some kind of catalyst used by the authors in this article: TaX_5_–Ph_4_Sn, (X = Cl, Br); TaBr_5_, CpCo(CO)_2_ − *hν*. The authors believe that there was a process of polycyclotrimerization.

Notably, [152] reported the synthesis of insoluble poly-m-DEB. In accordance with the IR spectra of the polymer, 1,2,4- and 1,3,5-substituted benzenes and unreacted –C≡CH groups are present in its structure. Unfortunately, the authors did not report on the polymerization conditions and the type of catalyst used.

Polymerization of m-DEB in the presence of [Rh(cod)ac] and [Rh(nbd)acac] ([M]_0_ = 0.6 mol L^−1^, [Cat]_0_ = 0.006 mol L^−1^, CH_2_Cl_2_, room temperature, 180 min,) gave in 2–3 min insoluble brown microporous polymers with yields of 90 and 83%, respectively [14]. For these polymers, the values S_BET_ = 498 and 653 m^2^ g^−1^, the volume of micropores 0.151 and 0.206 cm^3^ g^−1^, as well as the values of adsorbed hydrogen a(H_2_, 750torr) = 49.5 and 66.9 cm^3^ g^−1^, respectively, were obtained, which indicated a slightly lower microporosity of poly-m-DEB compared to poly-p-DEB. The polymer structure was determined using the methods of SEC and ^13^C CP/MAS NMR spectroscopy (Figure 14).

The original synthesis of π-conjugated micro/macroporous polyacetylene foams based on m-DEB is described in [16]. Initially, the π-conjugated foams were synthesized in the presence of the [Rh(nbd)acac] complex using deionized water and surfactant Span-80. As a result, with a yield of up to 66%, a solid brittle brown-red un-swelling (THF, CH_2_Cl_2_, CHCl_3_, benzene) polymer was obtained, representing polyene main chains that are crosslinked with 1,3-phenylene linkers (Figure 36).

The macropores walls were formed by microporous poly-m-DEB. The foam of the micro/microporous polymer contained about 0.6 unreacted terminal ethynyl group per one monomeric unit of the polymer skeleton. In the second stage, the foams were chemically modified using alkyne-azide click reaction between the lateral unreacted ethynyl groups of the polymer skeleton and D-glucose or cholesterol azides (Figure 36 and Figure 37), followed by thermal cross-linking at 280 °C. The S_BET_ value of the polymer increased from 110 to 380 m^2^ g^−1^ for the solid phase before and after hyperlinking, respectively.

According to the authors [16] micro/macroporous polyacetylene foams have great potential for expanding the application areas of this class of porous polymers.

The review [9] considers publications devoted to the use of rhodium catalysts in the polymerization of acetylene compounds, including for the polymerization of all DEB isomers. These publications are discussed above in more detail.

The original experiment is described in [13]. The authors conducted a one-pot self-encapsulation synthesis of a heterogeneous Pd catalyst based on m-DEB. According to the authors, the most effective ligand is Ph_3_P. The reaction consisted of mixing in one vial on the N_2_ medium m-DEB with the components of the catalyst Pd(OAc)_2_/Ph_3_P/methanesulfonic acid with the ratio Pd/PPh_3_/DEB = 1/6/10. During the synthesis, the formation of crosslinked polymer chains and encapsulation of the Pd catalyst occurred simultaneously with the formation of a heterogeneous Pd catalyst in the form of a black powder of an average size of 0.4 mm (Figure 38). Using XPS, it was found that encapsulated Pd species in the synthesized complex were present mainly in the form of Pd(0). The catalytic efficiency of the catalyst was evaluated in the air in the Suzuki-Miyaura, Stille, allylic arylation, and Mizoroki–Heck reactions. The high versatility and high catalytic activity of the Pd catalyst were found even with difficult reagents such as aryl chlorides and heteroaryl halogenides.

A palladium-based catalytic system (Pd(OAc)_2_/α,α-bis(di-t-butylphosphino)-o- xylene/methanesulfonic acid) was used to polymerize m-DEB [164] in a nitrogen atmosphere into the Schlenk flask for 18 h. Ultrasound was used to homogenize the system. The synthesized homo- and copolymers of m-DEB were used for their carbonization by heating to 800 °C in an N_2_ atmosphere. Poly-m-DEB, like other DEB monomers, had a very high carbonization yield of 83%, if carbonization was carried out without any carbonizing additives. For copolymers, the carbonization yields naturally decreased with a decrease in the amount of m-DEB in the copolymer. These carbonizates had a small surface area and low porosity, so their carbonization was carried out in the presence of KOH as a chemical activation agent. As a result, more microporous coals were obtained, in which the S_BET_ value increased from 450 to 784 m^2^ g^−1^. The authors believe that activated carbons are highly microporous with the presence of macropores and some mesopores having a narrow slit-like shape. Electrochemical results have shown excellent characteristics (high specific capacity, good stability, and low equivalent series resistance) of activated carbons. For example, the specific capacitance increased by more than 5 times from 35 to 181 F g^−1^ in the case of scanning at a speed of 25 mV s^−1^ with cyclic voltammetry. However, the best results were obtained for m-DEB copolymers with phenylacetylene. The results obtained allowed the authors [164] to recommend the use of such carbonisates as electrode materials in supercapacitors, as well also as sorbents for H_2_ storage and CO_2_ adsorption.

Using the SBA-15 template and m-DEB, a series of ordered mesoporous carbon materials with relatively high thermal stability was obtained [165]. The synthesis was carried out in accordance with the scheme shown in Figure 39. Notably, m-Diethylbenzene played the role of a carbon precursor. The samples differed in the final pyrolysis temperatures during synthesis (800 °C, 1000 °C, and 1200 °C). The synthesized carbon materials had a very narrow pore size distribution with a center of 4.3, 4.2, and 3.8 nm, respectively. In the case of the lowest carbonation temperature of the sample, the maximum values of the total pore volume and specific surface area (1.20 cm^3^·g^−1^ and 1044 m^2^·g^−1^, respectively) were obtained.

The paper [166] reports the polymerization of m-DEB using iCVD, which opens up possibilities for using polymers as dielectric insulators in electronic applications that cannot be combined with solvent-based materials. Moreover, iCVD is a solvent-free manufacturing method that allows poly-m-DEB layers to be deposited at a speed of 12 nm/min. The resulting polymers have a high molecular weight with a broad distribution (1000–25,000) and a well-defined Fourier transform infrared spectrum. The presence of a small peak at 2105 cm^−1^ and a monomer peak in the gel permeation chromatogram (GPC) suggests that some monomer is included in the layers.

Thus, only the use of the WCl_6_/phenylacetylene catalytic system can allow the synthesis of a soluble linear polymer m-DEB. However, in our opinion, this requires careful verification.

For a quick search for various methods of synthesis of DEB polymers, Appendix A summarizes information on synthesis methods and structural features of the resulting products.

## 5. Polymers of Polynuclear Diethynylarenes

In this section, diethynylarenes of the general formula HC≡C–R–C≡CH with various bridges R between ethynyl groups are considered. The considered monomers have no substituents in the aromatic fragments present in the bridges of R.

A new method for the synthesis of polyphenylenes using a catalytic reaction of polytricyclopolymerization of diacetylene-containing monomers was proposed in 1970 by the authors [66,67]. The HC≡C–Ar–C≡CH diethynylarenes’ structures were considered as monomers (Ar = –Ph–Ph–, –Ph–Ph–Ph–, –Ph–X–Ph– where X = O, S). It is proposed to use a catalytic system complex of the general formula [(RO)_3_P]_n_·CoHal, (where R = Alk_C≤6_, n = 1–4, Hal = Cl, Br, I). Based on the IR spectra of polymers, the same polymerization scheme for these monomers was proposed as for p-DEB (scheme in Figure 8), with the formation of crosslinked phenylene-containing polymers.

An insoluble polymer of 4,4′-diethynyldiphenyl (DEDP) was synthesized with a 57% conversion in the presence of the i-Bu_3_Al-TiCl_4_ complex [167]. According to the IR spectrum, the ratio of ethynyl groups to phenyl groups in the polymer was 1/10, and fragments of 1,2,4- and 1,3,5-substituted benzene were present in the structure of polymer chains.

DEDP polymerization was carried out in [18] in various ways. Polymerization in the presence of [Rh(cod)Cl]_2_ and [Rh(cod)im] (where cod = cis,cis-cyclo-octadiene; im = imidazole) yielded 100% and 90% insoluble brown polymers in 5 min and 12 h, respectively. Polymerization in the presence of Pd(PPh_3_)_2_Cl_2_] and [Pd(PPh_3_]_2_(DEDP)_2_] yielded only 85% and 50% of insoluble polymers in 20 and 24 h, respectively. Elemental analysis revealed a significant oxygen content in the polymers. There were uncertainties in the identification of signals in the IR spectra and XPS belonging to oxygen-containing groups. The authors suggested that the presence of oxygen may be associated with adsorbed water. In the IR spectra of polymers synthesized in the presence of complexes, bands at 3300 and 2100 cm^−1^ were observed, indicating the presence of groups –C≡CH. According to the authors, this is due to the presence of –p–Ph–p–Ph–C≡CH side groups in a rarely stitched polyDEDP. Heterogeneous doping of polymers was carried out by joint suspension of polyDEDP with FeCl_3_ or I_2_ (1/1 wt.) in THF, followed by drying and pressing of tablets. Doping made it possible to reduce the resistance of the polymer from ≈10^12^ Ω to R ≈ 6·10^5^ Ω and R ≈ 3·10^4^ Ω for FeCl_3_ and I_2_, respectively. It was found that Pt(II) complexes had practically no catalytic effect on the polymerization of the monomer.

Polymerization of DEDP [112] in the presence of ${\mathrm{Co}}_{2}{\left(\mathrm{CO}\right)}_{8}$ in anhydrous 1,4-dioxane under Ar at 125 °C for 1 h resulted in an insoluble polymer with 100% conversion. The absence of the C≡C–H bond vibrations at 3300 cm^−1^ indicated the complete depletion of ethynyl groups in the cyclopolymerization reaction (Figure 23). The polymer did not decompose in the air up to 360 °C but rather absorbed 0.22 wt. % H_2_ (60 kbar), had a total pore volume of 0.712 and a micropore volume of 0.341 cm^3^ g^−1^.

Boiling DEDP in a mixture of chloroform/paraffin oil (volume ratio 1/2) resulted in an insoluble brown polymer with a conversion rate of 90% in 2 h [18]. The IR spectrum showed the absence of bands at 3300 and 2100 cm^−1^ group –C≡CH, which indicated the implementation of a frequently crosslinked mesh in the polymer. Heterogeneous doping of the thermal polymer reduced the resistance of the polymer from R ≈ 10^12^ Ω to R ≈ 6·10^4^ Ω.

DEDP was proposed along with other diethynylarilenes to create potential thermosetting prepolymers [12] necessary for the creation of carbon/carbon composite materials.

A unique experiment to create a 2D mesh on an Au (111) substrate using DEDP is described in [19]. To do this, the DEDP molecules were thermally evaporated at 30 °C onto a purified gold substrate. Polymerization of alkynes was provided by subsequent annealing at 100 °C. The samples were examined using a scanning tunneling microscope with submolecular resolution and with using the density functional theory calculation. It was found that DEDP molecules form 2D networks on the Au (111) substrate due to a two-stage reaction [2+2+2]-cyclization of diyne on the surface of Au (111). According to the authors, they offer a way to create monatomic two-dimensional conjugate networks, the structure of which can be similar to graphene.

It was found that the homopolymerization of 2,5-diethynylthiophene creates various conjugated microporous polymers in two ways (Figure 40) [20]. The use of the [Rh(nbd)acac] complex in CH_2_Cl_2_ provided the formation of an insoluble non-swelling poly-conjugated polymer with a conversion of 100% in 1 h at 75 °C. At the same time, the main polymer chains were crosslinked by thiophene-2,5-diyl bridges (Figure 40A).

By changing the polymerization conditions (time and temperature), the authors [20] regulated the conversion of ethynyl groups in poly-conjugated networks and, at the same time, the degree of crosslinking of networks, which was proved by ^13^C CP/MAS NMR spectra. The use of the ${\mathrm{Co}}_{2}{\left(\mathrm{CO}\right)}_{8}$ complex in 1,4-dioxane provided polycyclotrimerization of the monomer with a conversion of 100% for 1 h at 120 °C (Figure 40B). The ^13^C CP/MAS NMR spectrum of the crosslinked polycyclotrimer revealed an almost quantitative transformation of the ethynyl groups of 2,5-diethynylthiophene during polycyclotrimerization with the formation of nodes of 1,2,4- and 1,3,5-substituted benzenes. The polymers synthesized using both complexes had SET values from 559 to 836 m^2^·g^−1^, micropore volumes from 0.6 to 0.23 cm^3^ g^−1^, and total pore volumes from 0.5 to 1.1 cm^3^ g^−1^.

The appearance of color in light even at room temperature in crystals of 1,4-diethyl naphthalene (1,4-DEN) and 1,4-diethyl-2,3-dichloronaphthalene [168] prompted the authors of [169,170] to investigate this process. It was found that the photo polymerization of 1,4-diethyl-2,3-dichloro-naphthalene proceeded slower compared to 1,4-DEN. The photopolymerization process was studied in sufficient detail on the example of 1,4-diethynylnaphthalene, for which several crystalline modifications of α, β, and γ with different reactivity were obtained. For the production of large quantities of polymers (poly-1,4-DEN), a suspension of the monomer and its deuterated derivatives was previously created in water or in mixtures of methanol and water. The suspensions were irradiated with UV light and then a synthesized soluble polymer with a conversion of 5–7% and a molecular weight of 9000 was extracted. It has been shown that the photoreactivity of 1,4-DEN is not due to the specific optimal packing of the monomer in crystals, as is the case with solid-phase topochemical polymerization of internal diacetylenes [2]. During storage, poly-1,4-DEN became insoluble. IR, resonant Raman, ^1^H NMR, and ESR spectra of polymers, including deuterated ones, were analyzed, and X-ray analysis was performed. The authors reported that “the ratios of reacted and unreacted ethynyl groups are close to one”. This allowed the authors to assert that “only one acetylene group per monomer participates in the polymerization reaction, and there is practically no crosslinking” [169,170]. The conclusions made by the authors admit the presence of branching in polyene polymer chains. Such uncertainty in the interpretation of the structure is indirectly confirmed by the fact that the $H_{\mathrm{ol}+\mathrm{ar}}/H_{\mathrm{eth}}$ proton ratios were not given.

Photopolymerization of the 1,4-DEN solution was carried out in cyclohexane under Ar with the formation of polymers and dimers [170]. The measurement of the molar absorption coefficient of the polymer proved the solubility of the polymer. Since the IR spectrum of the polymer was similar to the spectrum of the polymer synthesized by irradiation of crystals, this indicated a weak branching of polymer chains. The authors of [170] also reported the fact of polymer synthesis by the UV irradiation of a 1,4-diethyl naphthalene melt.

Irradiation of 1,4-DEN crystal samples by UV light (>330 nm) at 77 °K had no effect, although a radical polymerization reaction was initiated. After heating the samples to −15 °C without further irradiation, their color changed in 30 s from white to bright yellow and then to dark red, which was explained by the appearance of longer polyconjoined chains [170].

Thermal polymerization of 1,4-diethyl-2,3-dichloro-naphthalene was more effective than 1,4-DEN (Ar medium, thermostat). The polymers were insoluble, but a slight weakening of the band at 3300 cm^−1^ indicated a small number of crosslinking due to the opening of the lateral groups –C≡CH [170].

1,8-diethinyl naphthalene (1,8-DEN) was polymerized using various transition metal-based catalysts with a conversion of 56–97% at 70 °C in 24 h [171]. In the polymer (poly-1,8-DEN) synthesized in the presence of MoCl_5_ or WCl_6_, the soluble part was only 20%. Given the existence of an insoluble fraction, there is a high probability of branching even in soluble polymer molecules. The polymer obtained in the presence of PdCl_2_ is readily soluble in CHCl_3_, THF, and DMSO and had ${\bar{M}}_{n}\;$= 2.8·10^3^. The solubility of polymers synthesized in the presence of MoCl_5_/EtAlCl_2_ (1/4 mol/mol) and WCl_6_/(n-Bu)_4_Sn (1/4 mol/mol) has not been reported. In the ^1^H-NMR spectrum of poly-1,8-DEN, there was no ethynyl proton singlet. In the IR spectrum, there were no vibrations with frequencies of 3287 and 2113 cm^−1^ that were responsible for fluctuations in the C≡C–H and –C≡C– bonds of ethynyl groups. This allowed the authors to propose a probable polymerization scheme due to the disclosure of groups –C≡C–H without taking into account possible ramifications and without specifying the reasons for the insolubility of some of the synthesized polymers (Figure 41).

Unfortunately, the authors did not indicate which catalysts were used to synthesize polymers, the spectra of which were studied and shown in the figures.

Hyperbranched crosslinked polycyclotrimers of 2,6-diethinyl naphthalene (2,6-DEN) and 2,6-diethinylanthracene (2,6-DEA) were synthesized in the presence of the TaCl_5_/Ph_4_Sn complex [116]. The reaction was carried out at benzene at room temperature for 24 h. As a result, insoluble orange poly-2,6-DEN and poly-2,6-DEA were obtained with a conversion rate of 81 and 74%, respectively. ^13^C-CP/MAS NMR and IR spectra demonstrated the presence in the polymer structure of a certain number of untransformed groups –C≡C–H and crosslinking linkers in the form of 1,2,4- or 1,3,5-substituted benzene, as in the case of poly-p-DEB (Figure 24). The sorption capacity of these polymers was compared with the sorption capacity of poly-p-DEB. The S_BET_ values were 1299, 418, and 9 m^2^ g^−1^ respectively for poly-n-DEB, poly-2,6-DEN, and poly-2,6-DEA. Naturally, the adsorption capacity for N_2_ decreased in the same order for these polymers in the range 51.0 > 6.6 > 0.75 (mmol N_2_)·g^−1^.

Notably, 4,4′-Diethynyldiphenylmethane has been proposed along with other diethynylenes to create potential thermosetting prepolymers [12] necessary for the creation of carbon/carbon composite materials.

A unique result was obtained in [172]. In order to reduce the fragility of the matrix and increase the adhesion of the matrix to the fibers, the authors proposed a 4,4′-diethynyldiphenyloxide monomer having a hinged oxygen atom as the basis of the matrix. Thermal polymerization of the monomer in the melt began at a temperature of 150 °C. A series of samples obtained in the range of increasing temperatures of 180–300 °C was studied using a number of physicochemical NMR, IR, and DSC methods. This allowed the authors to detect a consistent decrease in the number of acetylene protons in polymers with a simultaneous increase in the number of vinyl hydrogen atoms. At the same time, the number of aromatic atoms did not increase, which indicated the absence of cyclopolymerization. According to the authors, the polymer structure is a substituted cis-polyene with a possible number of branches –[C=C–(Ph–O–Ph–C≡CH)]_n_–. This result is the only example of selective disclosure of only one bond –C≡C– during the thermal polymerization of a diethynyl monomer.

A ${\left[{\left(C_{2}H_{5}O\right)}_{3}P\right]}_{4}\cdot \mathrm{CoBr}$ catalyst has been proposed to obtain branched crosslinked polyphenylene-type polymers using a polycyclotrimerization reaction of various diethynylarylene monomers [66]. However, only one example describes the polymerization of 4,4′-diethynyldiphenyloxide at a temperature of 75–78 °C for 6 h. An insoluble polymer was obtained with a yield of 10%.

The use of the i-Bu_3_Al-TiCl_4_ complex as a catalyst made it possible to synthesize, with a yield of 70%, only an insoluble polymer of 4,4′-diethynyldiphenyloxide having (according to IR spectroscopy results) 1,2,4- and 1,3,5-substituted benzene in the structure [167]. At the same time, the ratio of ethynyl groups to phenyl groups was 1/30.

In [173], another organosilicon spacer was proposed to reduce the overall stiffness of a polymer with the general formula HC≡C–Ph–Si(C_n_H_2n+1_)_2_–Ph–C≡CH (n = 2, 4, 6). It is very interesting that the authors managed to synthesize a series of hyperbranched but soluble polysilylenephenylenes (PSP) using a TaBr_5_ catalyst. It was found that polymerization took place only in the toluene medium and was absent in DO, THF, CH_2_Cl_2_, and hexane. The effect of the Ph_4_Sn co-catalyst, the concentration of the catalyst and monomer, the time and temperature of polymerization on the conversion, and the molecular weight of the polymer were studied. In particular, it was possible to synthesize PSP with a conversion of 100%, ${\bar{M}}_{w}\;$= 19,700 and ${\bar{M}}_{w}/{\bar{M}}_{n}$ = 2.5 under the following synthesis conditions: [Cat] = 10 mM, [M]_0_ = 0.1 M, N_2_ medium, room temperature, and 24 h. In the IR spectra of polymers, an almost complete disappearance of the vibrational frequencies of the ≡C–H and C≡CH bonds was detected with a simultaneous increase in the absorption band of the aromatic skeleton C=C at 1594 cm^−1^. This proved the formation of a new number of 1,3,5- and 1,2,4-substituted benzene rings. The combined use of ^1^H and ^13^C NMR, IR spectra, and computer modeling allowed the authors to propose the following PSP structure (Figure 42).

The authors of [173] demonstrated that PSPs are easily photocrosslinked, thereby producing good-resolution fluorescent photographic images.

A series of four monomers with the general formula HC≡C–R–C≡CH (R=–Ph–C(Ph)=C(Ph)–Ph–; –Ph–O–(CH_2_)_6_–O–Ph–; –CH_2_–O–Ph–CH_2_–Ph–O–CH_2_–; –C(O)–Ph–O–(CH_2_)_6_–O–Ph–C(O)–) was used to create soluble, regioregular hyperbranched polymers [174]. Polycyclotrimerization of monomers was carried out using the catalyst InCl_3_/2-iodophenol in chlorobenzene at 130 °C under N_2_. At concentrations [M]_0_= 0.15 M, [In] = 0.011 or 0.015 M, [2-iodophenol] = 0.10 M, soluble polymers with a conversion of 50.0–94.1%, ${\bar{M}}_{w}$ = 5400–13,500 and a ${\bar{M}}_{w}/{\bar{M}}_{n}\;$= 1.54–2.68 were synthesized in 2 h. In the IR spectra of all polymers, there were no absorption bands at 3305 and 2102 cm^−1^, due to the stretching of ≡CH and C≡C, respectively, and characteristic of the initial monomers. The ^1^H- and ^13^C-NMR spectra of the polymers showed that only a derivative of 1,3,5-regioregular benzene is formed (Figure 43).

For a quick search for various methods of synthesis of polynuclear diethynylarenes polymers, Appendix A summarizes information on synthesis methods and structural features of the resulting products.

## 6. Conclusions

Homopolymerization of bifunctional diethynylarenes can proceed both along one and both bonds –C≡C- with the formation of polymers of three different types of intramolecular structure: linear, branched, and crosslinked. The type of intramolecular structure formed is influenced by the polymerization conditions.

Solid-phase polymerization of crystalline diethynylarenes leads to polymers of various molecular weights, including insoluble ones, with virtually no established intramolecular structure.

Gas-phase polymerization and liquid-phase polymerization of diethynylarenes without the use of initiators lead to insoluble, highly crosslinked polymers. In a number of these papers, the authors propose a possible (most likely) structure.

The use of various initiating systems in the case of liquid-phase polymerization of diethynylarenes leads, as a rule, to the synthesis of insoluble and crosslinked products, as well as to the synthesis of soluble and branched polymers. In this case, the type of initiating system significantly affects the intramolecular structure of the resulting polymers. One group of catalysts provides a synthesis of crosslinked or branched polymers of polyene structures. Another group of catalysts promotes the formation of 1, 2, 4- and 1, 3, 5-phenylene fragments in both branched soluble polymers and crosslinked polymers. The latter fact is natural since the use of these catalysts makes it possible to obtain 1, 2, 4- and 1, 3, 5- substituted benzene.

A very limited number of articles describe the synthesis of soluble polymers having a complete polyene unbranched structure. It is on the basis of these polymers that it is possible to obtain polymer clusters or grafted copolymers using polymer-analogous transformations or click reactions, which will make it possible to create materials of a new type. However, for these purposes, such polymers should not have intramolecular defects formed due to the existence of various cis-trans isomers, as well as due to the possible addition of polymer units of the head-tail or head-head type.

In the vast majority of publications, the authors describe the complex physicochemical properties as well as the areas of application of the corresponding polymers, taking into account their properties and features of the intramolecular structure.

## Figures and Tables

**Figure 1 polymers-15-01105-f001:**
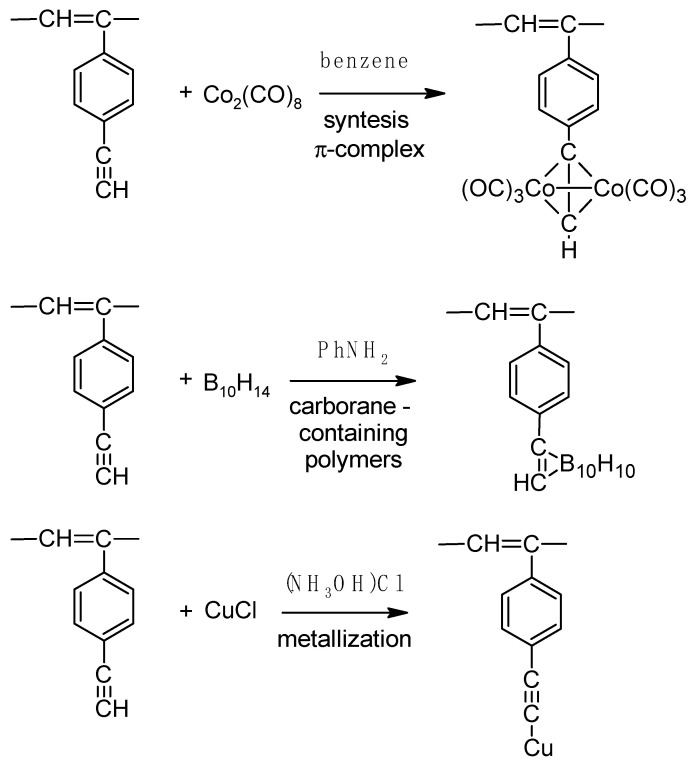
Schemes of the synthesis of organic element derivatives of poly-p-DEB.

**Figure 2 polymers-15-01105-f002:**
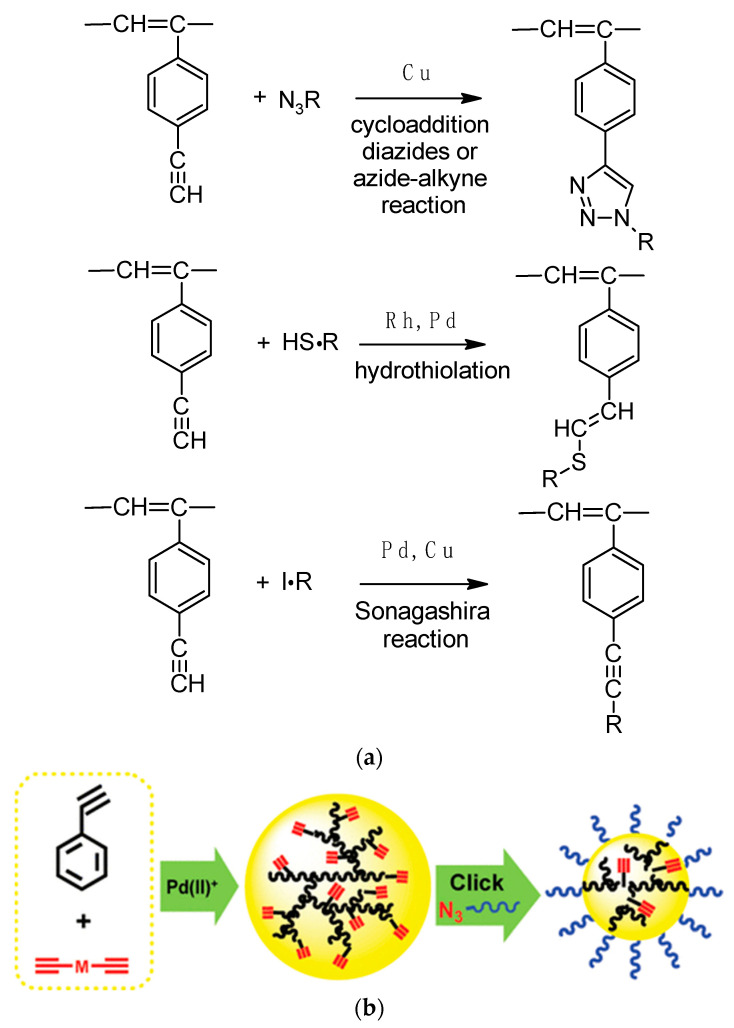
(**a**) Schemes of the possible click reactions synthesis of p-DEB polymers, where R is substituents of benzene, pyridine, thiophene, carbazole, etc. (**b**) The azide-alkyne click reaction scheme from [39]. Reprinted with permission from [39]. Copyright 2012, American Chemical Society.

**Figure 3 polymers-15-01105-f003:**
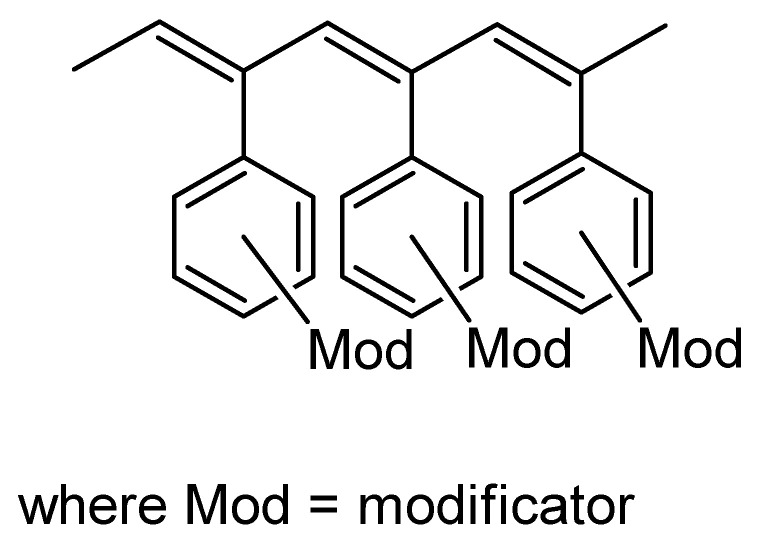
Probable structure of a polymer cluster based on linear trans-S-trans poly-DEB.

**Figure 4 polymers-15-01105-f004:**
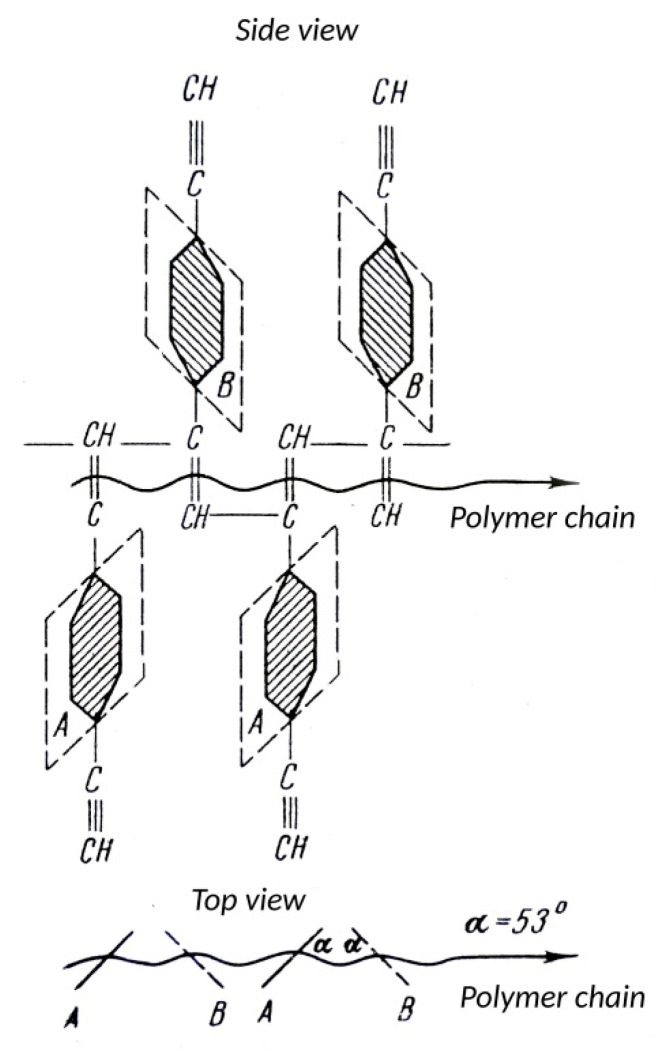
The scheme of possible poly-p-DEB grew from article [41] with translated captures. Reprinted with permission from Broude, V.L. (1968). Copyright 1968 Institute of Chemical Physics RAS.

**Figure 5 polymers-15-01105-f005:**
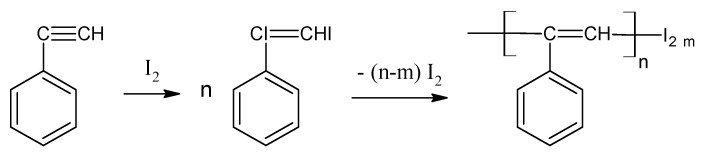
The scheme of dehalopolycondensation. Reprinted with permission from Ref. [63] Copyright 1989 Elsevier.

**Figure 6 polymers-15-01105-f006:**
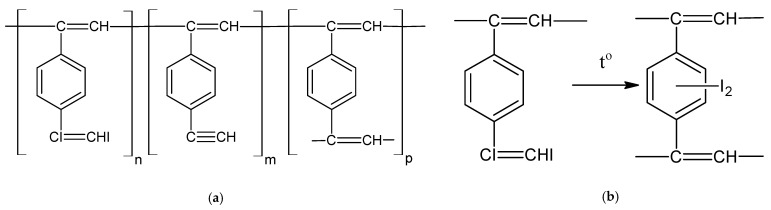
(**a**) The schemes of p-diethinylbenzene structure, where n >> m,p; (**b**) The schemes of polymer complex formation. Reprinted with permission from Ref. [63] Copyright 1989 Elsevier.

**Figure 7 polymers-15-01105-f007:**
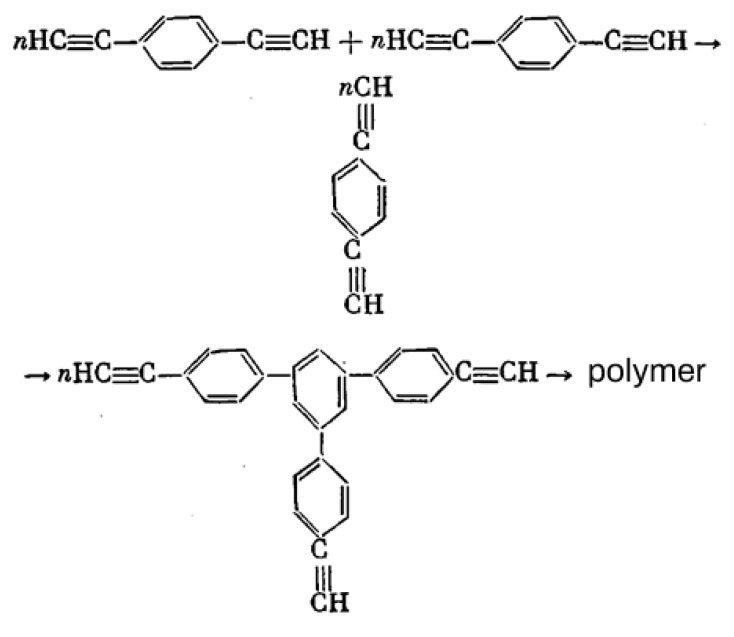
The schemes of polymer mesh formation from [70] with translated captures. Reprinted with permission from Korshak, V.V. (1972). Copyright 1972 Institute of Organoelement Compounds RAS.

**Figure 8 polymers-15-01105-f008:**
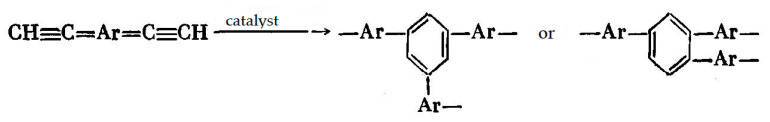
The schemes of the mechanism of polycyclotrimerization with the formation of 1,3,5- and 1,2,4 (98%)-substituted phenylene fragments from [74] with translated captures. Reprinted with permission from Korshak, V.V. (1971). Copyright 1971 Institute of Organoelement Compounds RAS.

**Figure 9 polymers-15-01105-f009:**
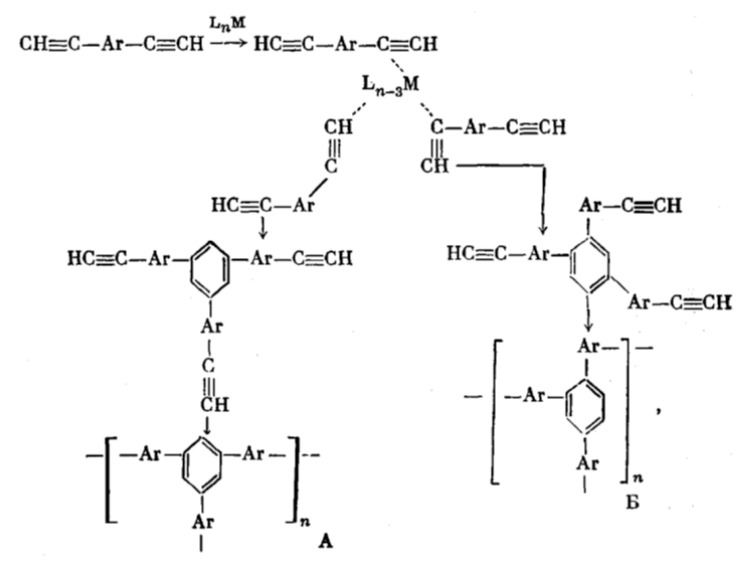
The schemes the formation of a complex of three ethynyl groups with a transition metal atom from [75], where M is transition metal atom, L is ligands. Reprinted with permission from Korshak, V.V. (1973). Copyright 1973 Institute of Organoelement Compounds RAS.

**Figure 10 polymers-15-01105-f010:**
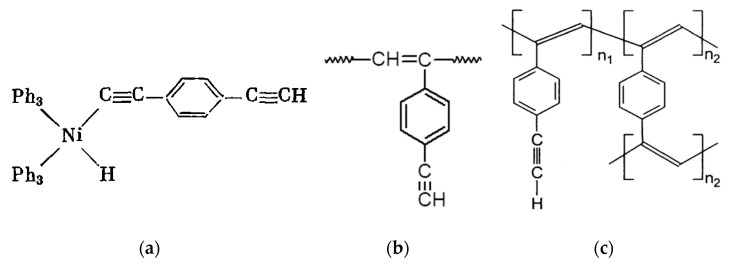
(**a**) intermediate active center—hydride acetylenide (**b**) linear structure of non-branched polyene in poly-p-DEB (**c**) trans-polyene structure with a certain number of side groups.

**Figure 11 polymers-15-01105-f011:**
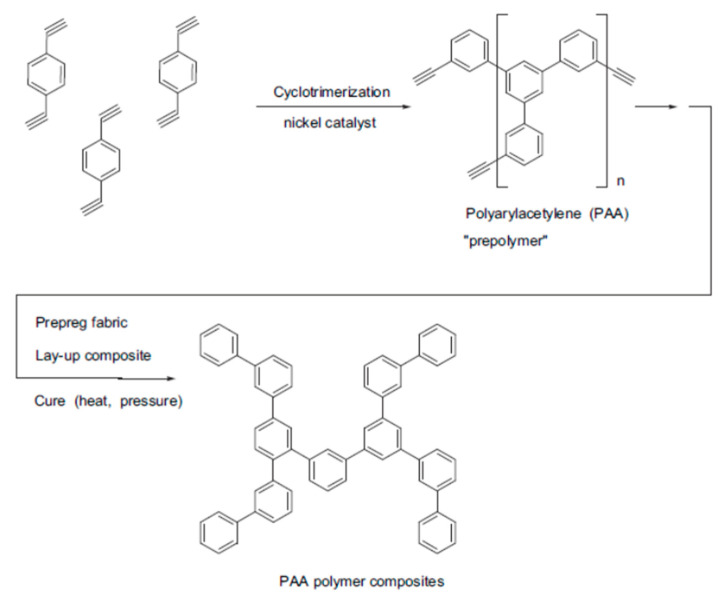
Synthesis of cyclotrimerized PAA prepolymer and subsequent processing steps to produce PAA polymer composites. Reprinted with permission from [64] Copyright 2009 Elsevier Ltd.

**Figure 12 polymers-15-01105-f012:**
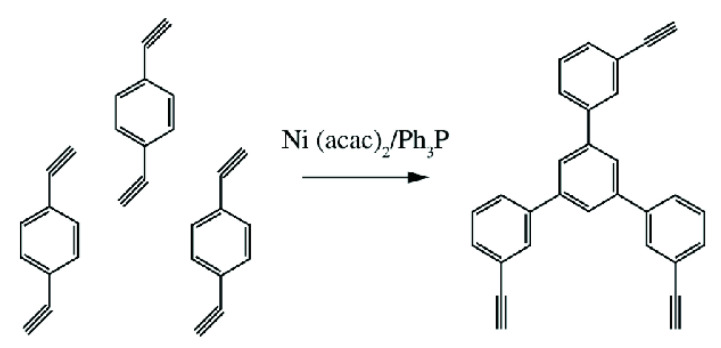
Cyclotrimerization reaction of 1,4-diethynylbenzene. Reprinted with permission from Oishi, S.S. (2014). Copyright 2014 Oishi S.S.

**Figure 13 polymers-15-01105-f013:**
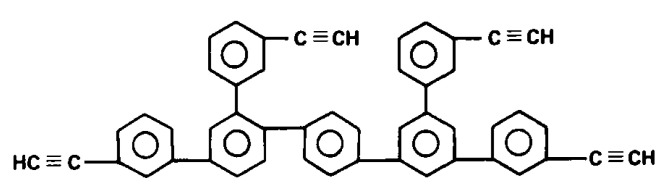
The structure of p-DEB oligomers. Reprinted with permission from [23]. Copyright 2007 Taylor & Francis.

**Figure 14 polymers-15-01105-f014:**
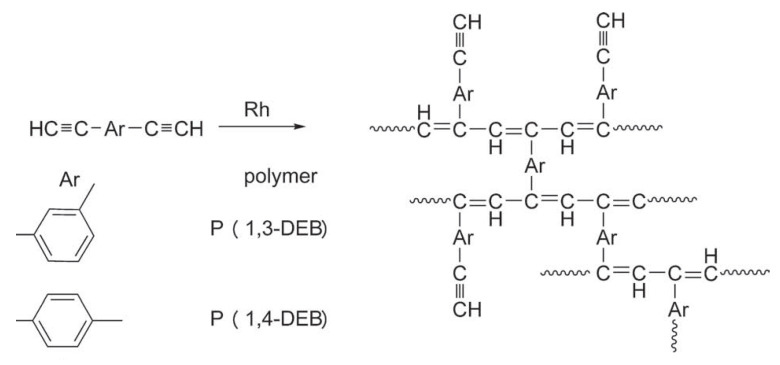
Synthesis of polyacetylene-type microporous organic polymers. Reprinted with permission from [14]. Copyright 2011 John Wiley and Sons, Inc.

**Figure 15 polymers-15-01105-f015:**
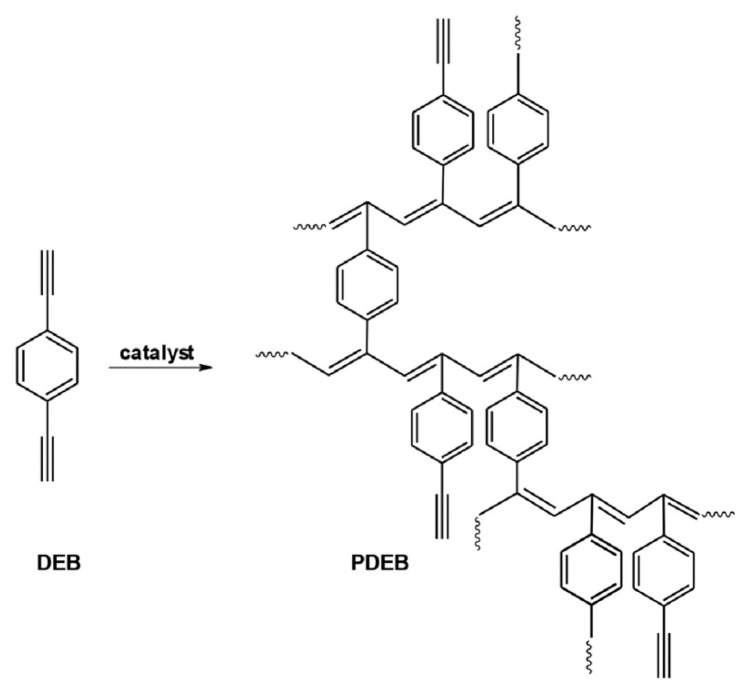
Chain-growth polymerization of DEB with transition metal catalysts. Reprinted with permission [15]. Copyright 2014 John Wiley and Sons, Inc.

**Figure 16 polymers-15-01105-f016:**
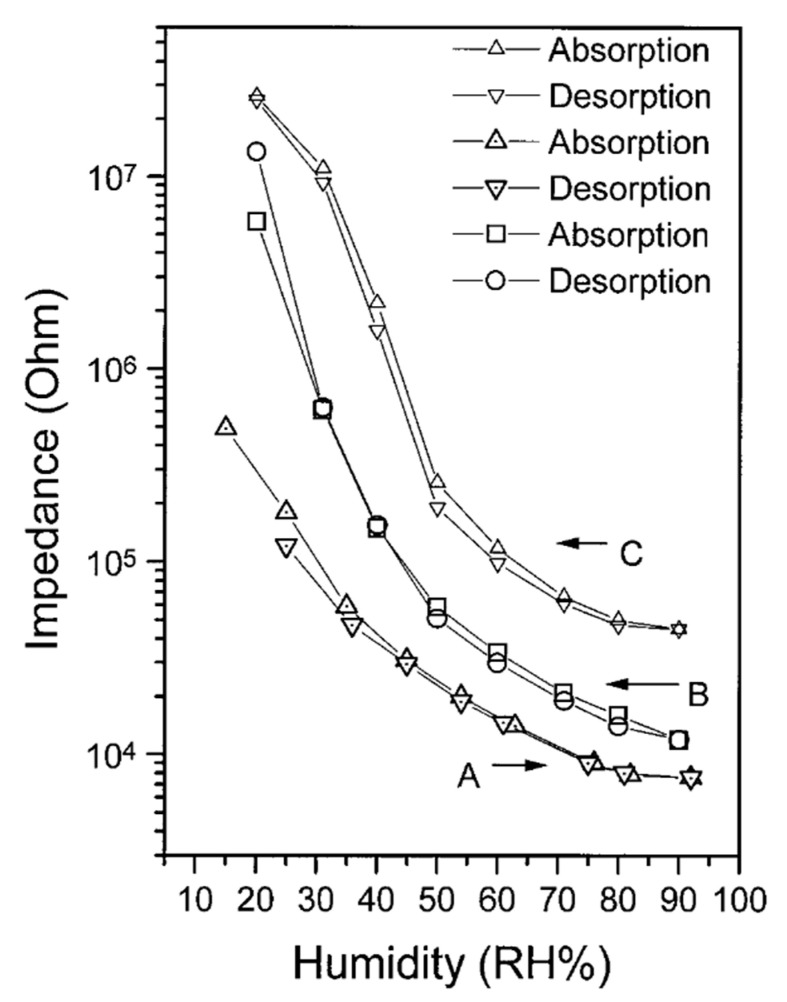
Humidity response of PDEB-based sensors (A, B, and C) prepared with different methods. Reprinted with permission from Ref. [95]. Copyright 1999 John Wiley and Sons, Inc.

**Figure 17 polymers-15-01105-f017:**
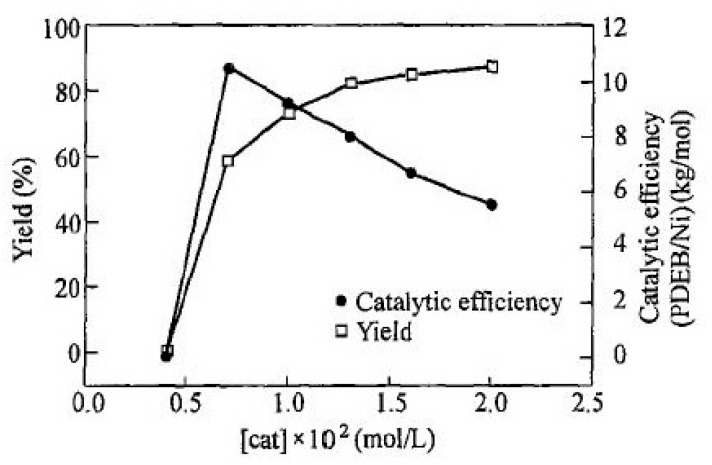
Effect of catalyst concentration. Conditions: volume ratio of 1,4-dioxane/toluene = 1, [M] = 1.0 mol/L, 25 °C, 6 h. Reprinted with permission from [98]. Copyright 2001 Springer Nature.

**Figure 18 polymers-15-01105-f018:**
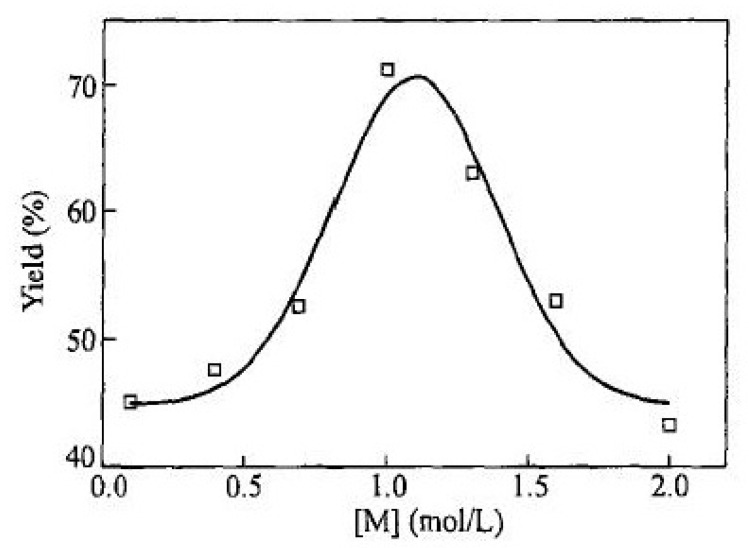
Effect of monomer concentration. Conditions: volume ratio of 1,4-dioxane/toluene = 1, [cat] = 0.01 mol/L, 25 °C, 6 h. Reprinted with permission from [98]. Copyright 2001 Springer Nature.

**Figure 19 polymers-15-01105-f019:**
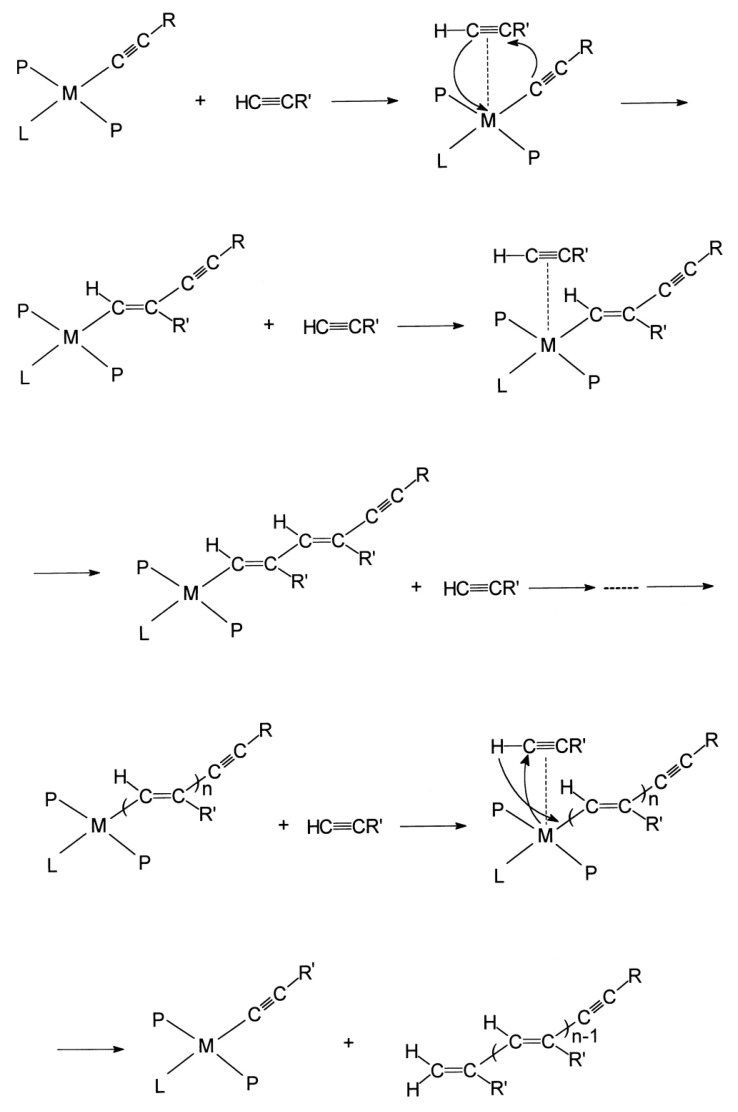
Polymerization mechanism of alkynes by transition metal acetylide catalysts. P: PPh_3_, PBu_3_, Ph_2_PCH_2_CH_2_PPh_2_; L: C≡CR, Cl; M: Ni, Pd. Reprinted with permission from [101]. Copyright 2001 Elsevier.

**Figure 20 polymers-15-01105-f020:**
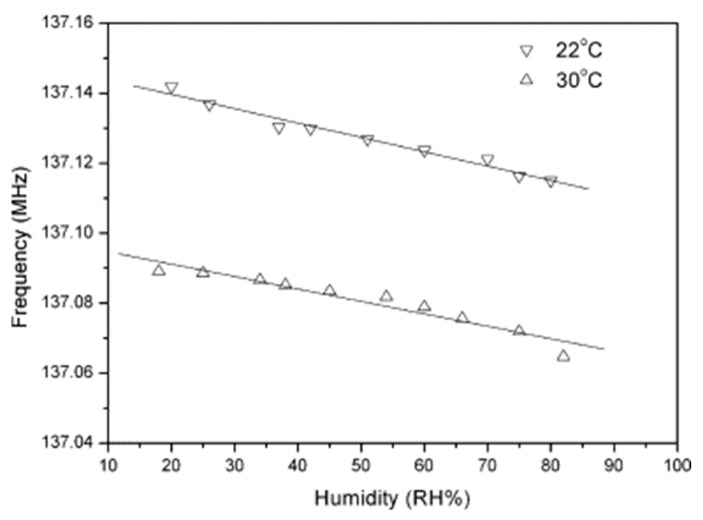
The frequency response to the relative humidity of the SAW sensor based on PDEB at different temperatures. Reprinted with permission from [103]. Copyright 2007 Elsevier.

**Figure 21 polymers-15-01105-f021:**
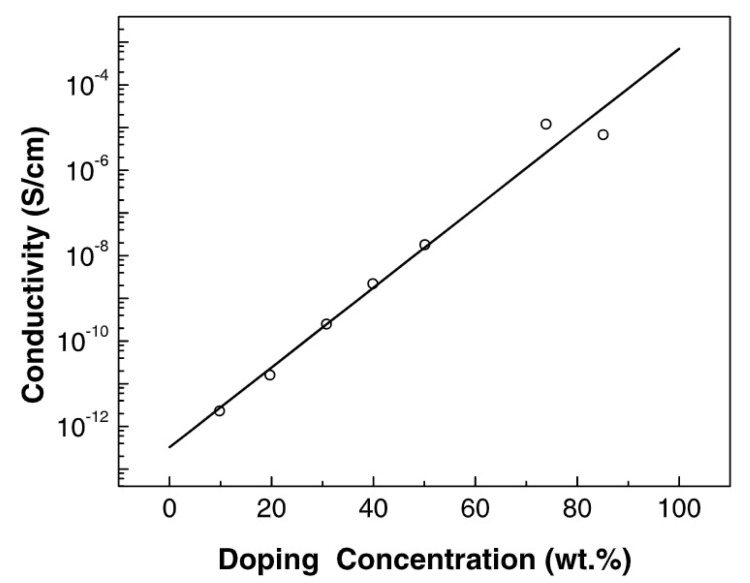
Conductivity as a function of iodine doping concentration. Reprinted with permission from [106]. Copyright 2002 Elsevier.

**Figure 22 polymers-15-01105-f022:**
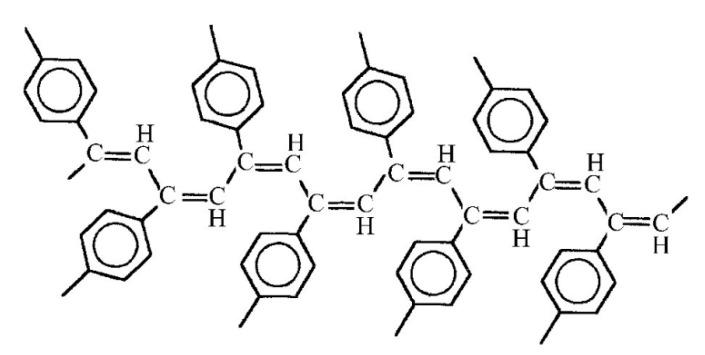
Structure of cis-polyene without aromatic fragments. Reprinted with permission from [107]. Copyright 2001 East China University of Science and Technology.

**Figure 23 polymers-15-01105-f023:**
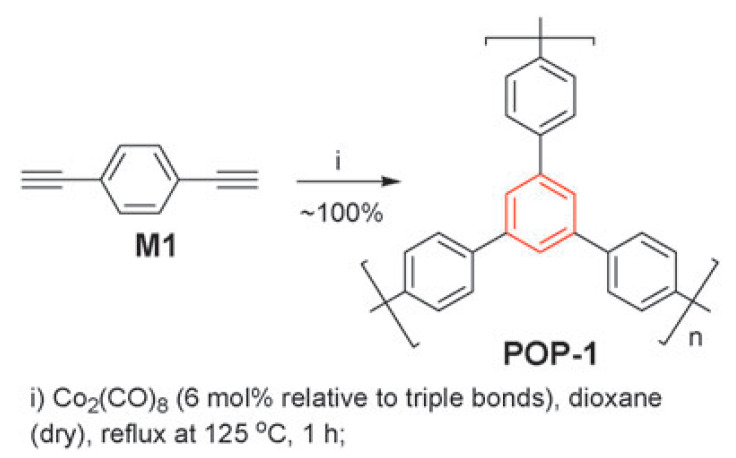
Preparation of POP-1 (benzene in red indicates one of the potential trimerization pathways from ethynyl groups). Reprinted with permission from [112]. Copyright 2010 The Royal Society of Chemistry.

**Figure 24 polymers-15-01105-f024:**
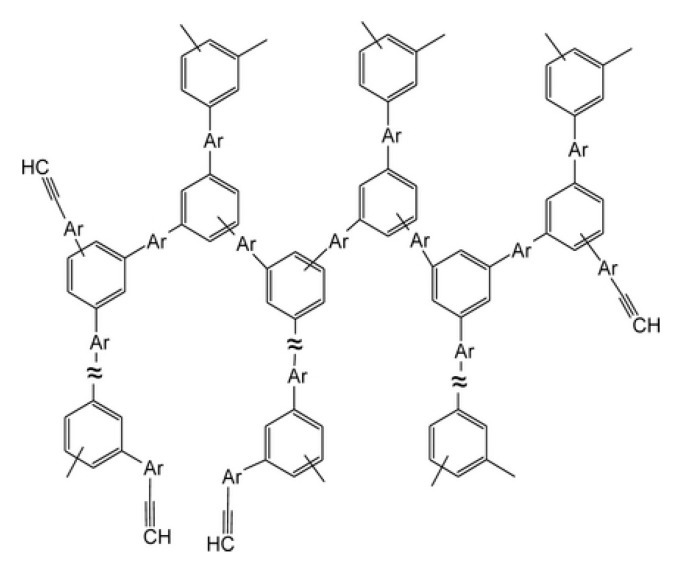
The scheme of polycyclotrimers of diethynylarenes. Reprinted with permission from [116]. Copyright 2013 John Wiley and Sons, Inc.

**Figure 25 polymers-15-01105-f025:**
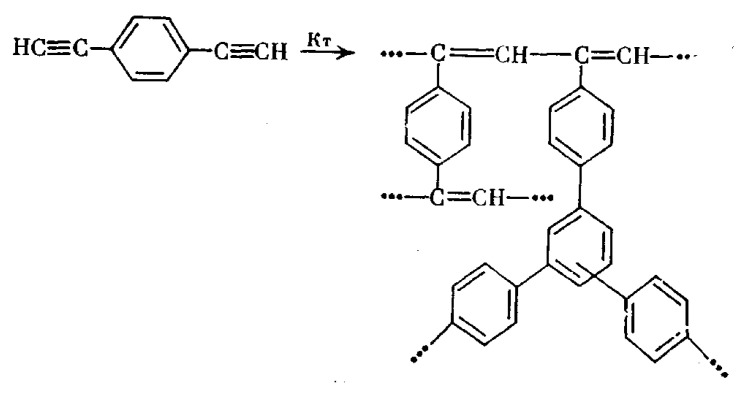
The scheme of polymerization of p-DEB, where K_T_ is a catalyst. Reprinted with permission from Sergeev, V.A. (1977). Copyright 1977 Institute of Organoelement Compounds RAS.

**Figure 26 polymers-15-01105-f026:**
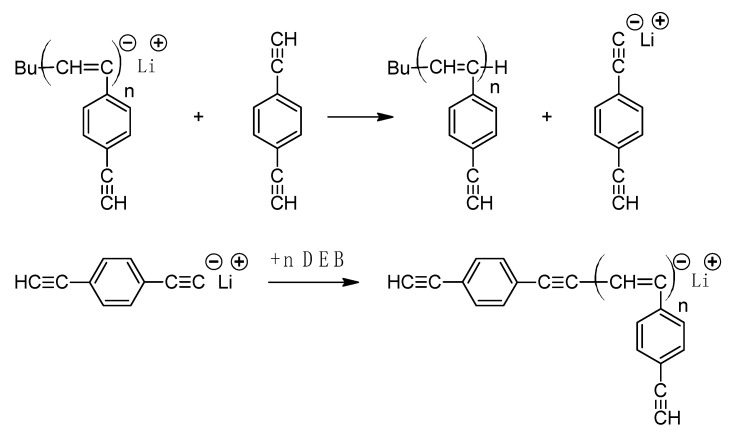
The scheme of chain transfer reaction to p-DEB. Reprinted from Ref. [33].

**Figure 27 polymers-15-01105-f027:**
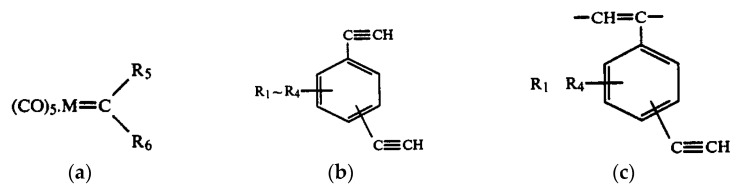
(**a**) carbenes structure (**b**) diethynylbenzenes structure (**c**) linear polymer of p-DEB structure.

**Figure 28 polymers-15-01105-f028:**
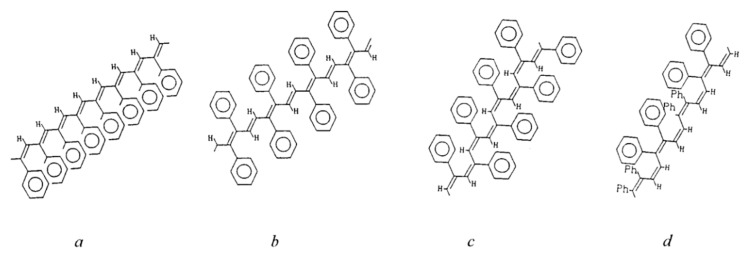
Steric hindrances in PPA chains: planar projections of PPA chains formed in the metathesis polymerization: (**a**) HT trans-transoid chain; (**b**) HH-TT trans-transoid chain; (**c**) HT cis-transoid chain; (**d**) HH-TT cis-transoid chain. Reprinted with permission from [141]. Copyright 1999, American Chemical Society.

**Figure 29 polymers-15-01105-f029:**
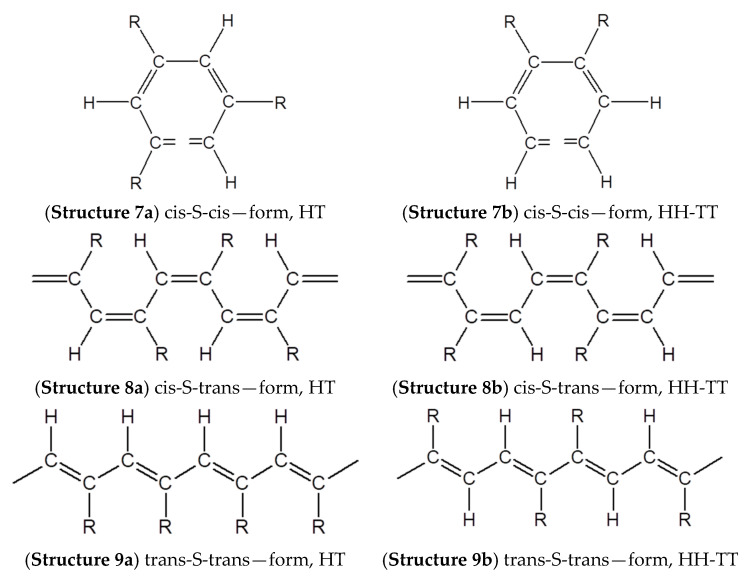
Possible conformal structures of disubstituted polyenes, where R = –PhC≡CH [33].

**Figure 30 polymers-15-01105-f030:**
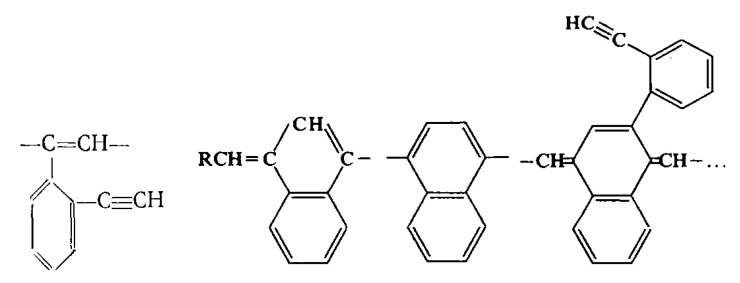
Possible poly-o-DEB fragments. Reprinted with permission from [145]. Copyright 2003, John Wiley and Sons, Inc.

**Figure 31 polymers-15-01105-f031:**
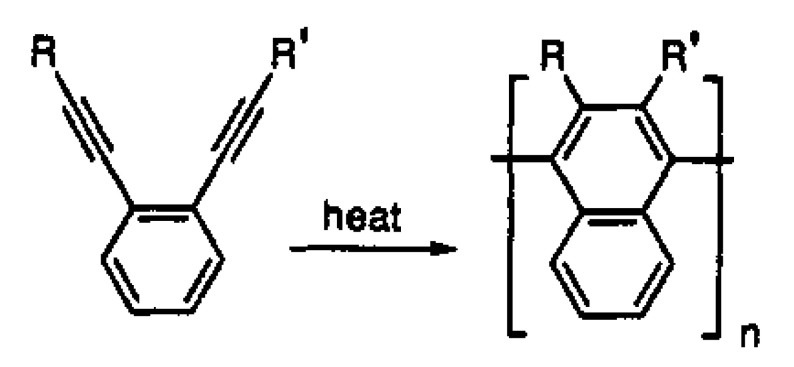
Scheme of thermal polymerization of o-DEB in accordance with [146,147]. Reprinted with permission from [146,147]. Copyright 1997, Elsevier.

**Figure 32 polymers-15-01105-f032:**
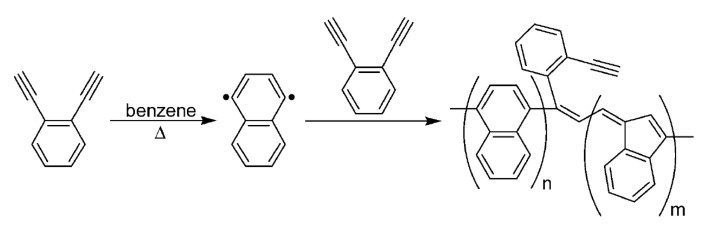
Scheme of thermal polymerization of o-DEB. Reprinted with permission [148]. Copyright 2003, American Chemical Society.

**Figure 33 polymers-15-01105-f033:**
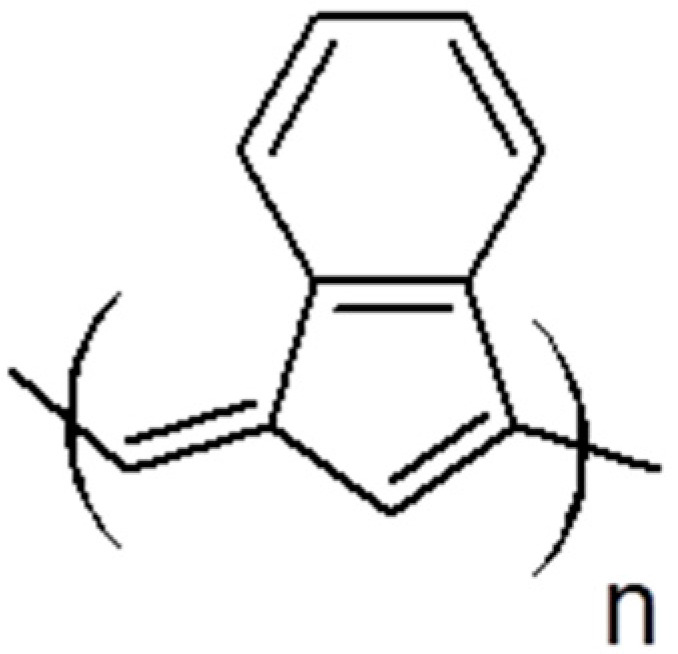
Structure of poly-o-DEB synthesized in the presence of metathesis catalysts. Reprinted from Ref. [149].

**Figure 34 polymers-15-01105-f034:**
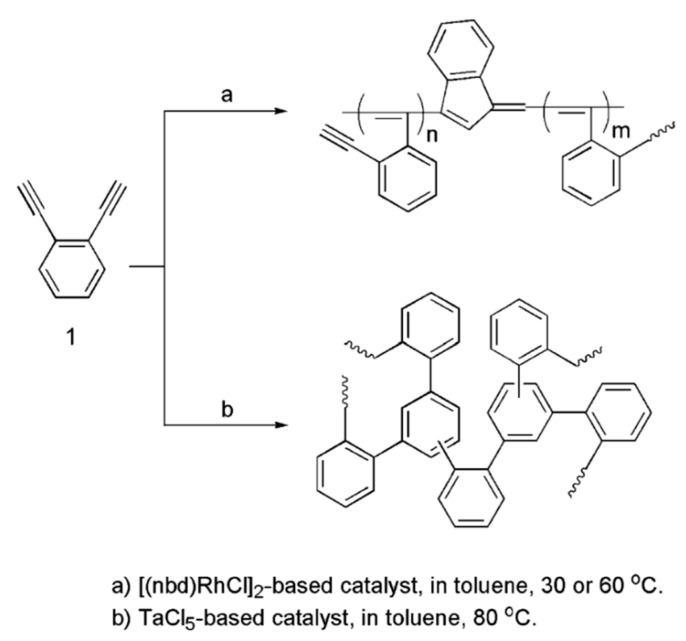
Structure of poly-o-DEB synthesized. Reprinted with permission from [17]. Copyright 2006, Elsevier.

**Figure 35 polymers-15-01105-f035:**
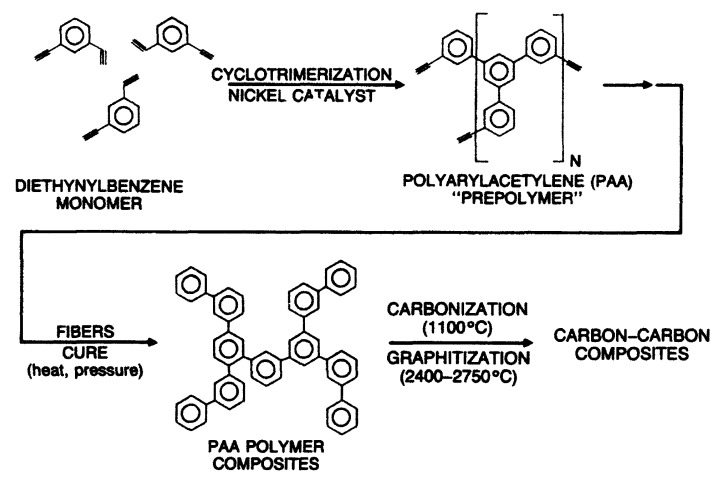
Scheme of polycyclotrimerization of m-DEB. Reprinted with permission from [154,155,156]. Copyright 1991, Elsevier.

**Figure 36 polymers-15-01105-f036:**
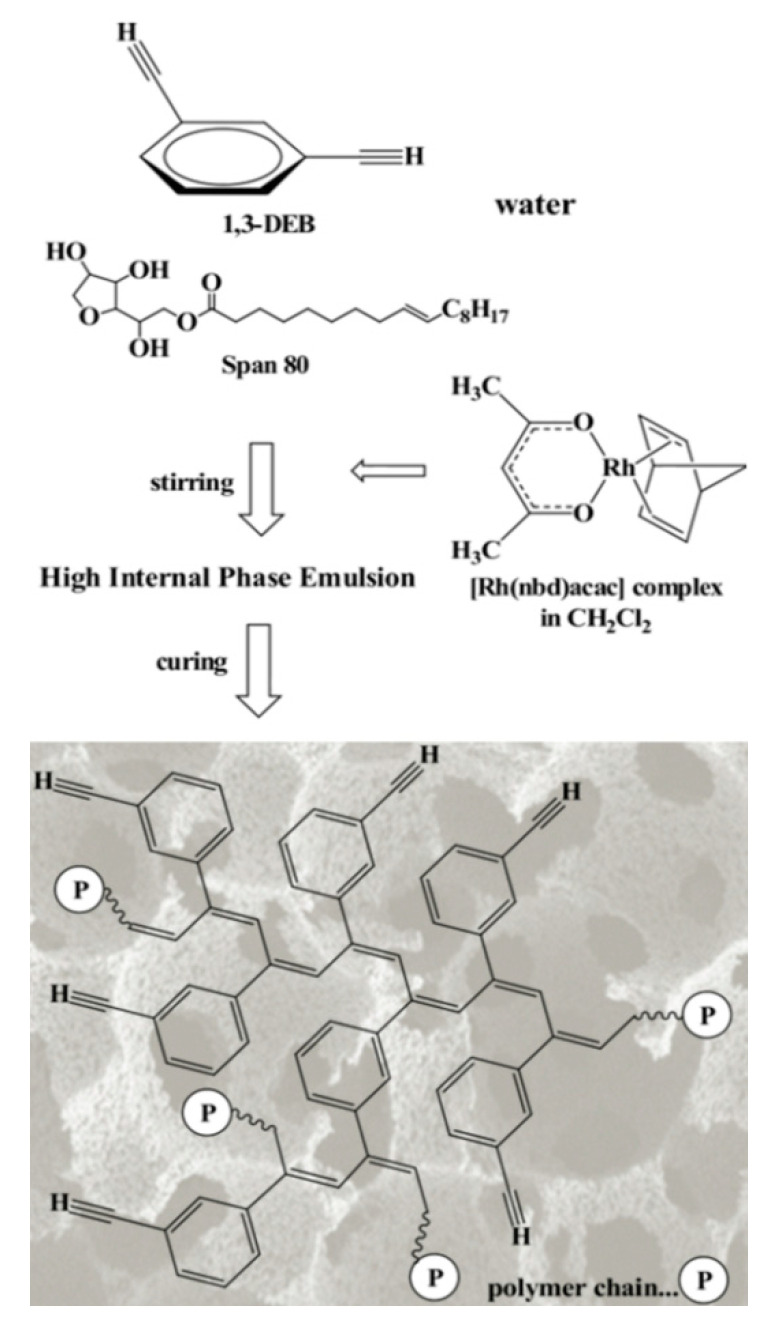
Scheme of preparation protocol of the π-Conjugated PolyHIPE Foams with Intrinsic Porosity. Reprinted with permission from [16]. Copyright 2014, American Chemical Society.

**Figure 37 polymers-15-01105-f037:**
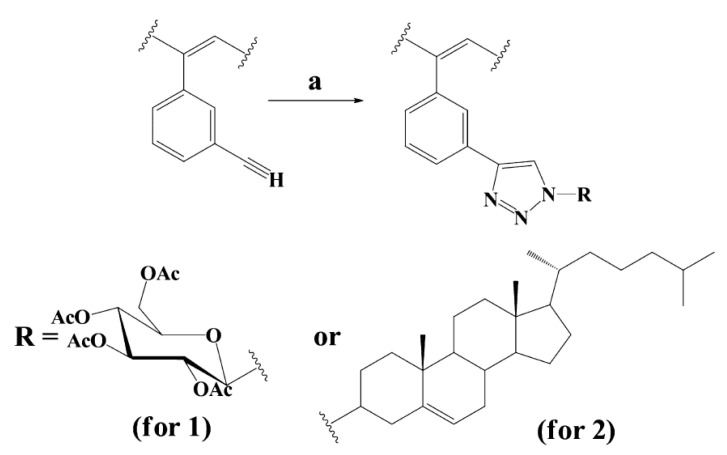
Chemical Modification Reactions: RN3, Heating in (a) Toluene (for 1) or Xylene (for 2). Reprinted with permission from [16]. Copyright 2014, American Chemical Society.

**Figure 38 polymers-15-01105-f038:**
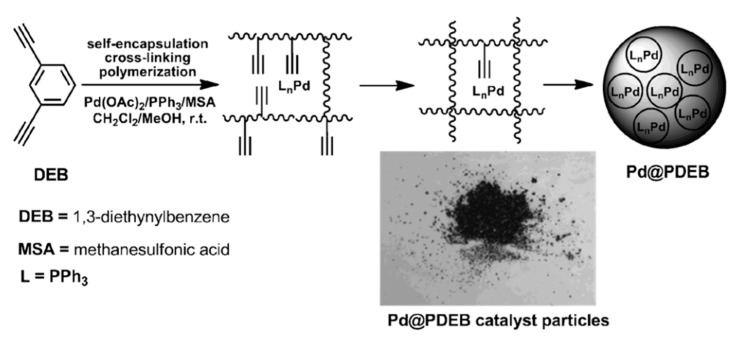
One-step synthesis of heterogeneous Pd@PDEB via self-encapsulation cross-linking polymerization of 1,3-diethynylbenzenes. Reprinted with permission from [13]. Copyright 2014, John Wiley and Sons, Inc.

**Figure 39 polymers-15-01105-f039:**
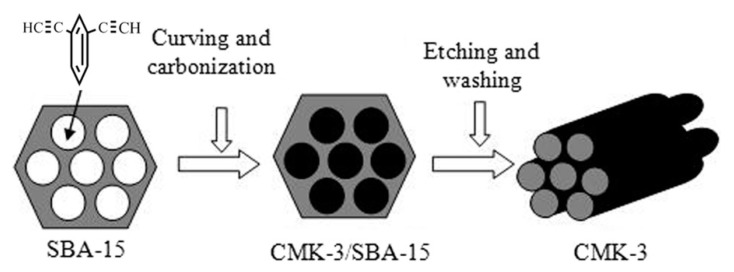
Scheme of synthesis of mesoporous carbon materials. Reprinted with permission from [165]. Copyright 2017, Elsevier.

**Figure 40 polymers-15-01105-f040:**
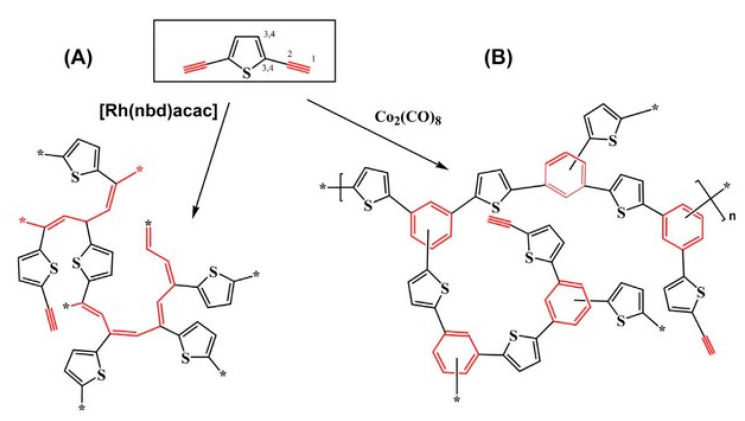
Homopolymerization of DET into (**A**) polyacetylene network and (**B**) polycyclotrimer network. * is continuation of the polymer. Reprinted with permission from [20]. Copyright 2017, Elsevier.

**Figure 41 polymers-15-01105-f041:**
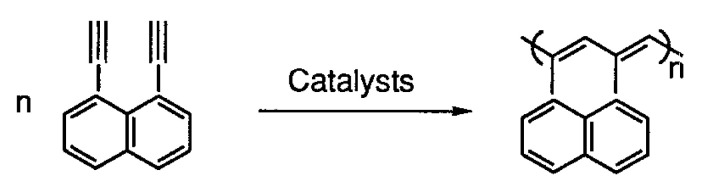
Scheme of cyclopolymerization of DEN. Reprinted with permission from [171]. Copyright 1995, Springer Nature.

**Figure 42 polymers-15-01105-f042:**
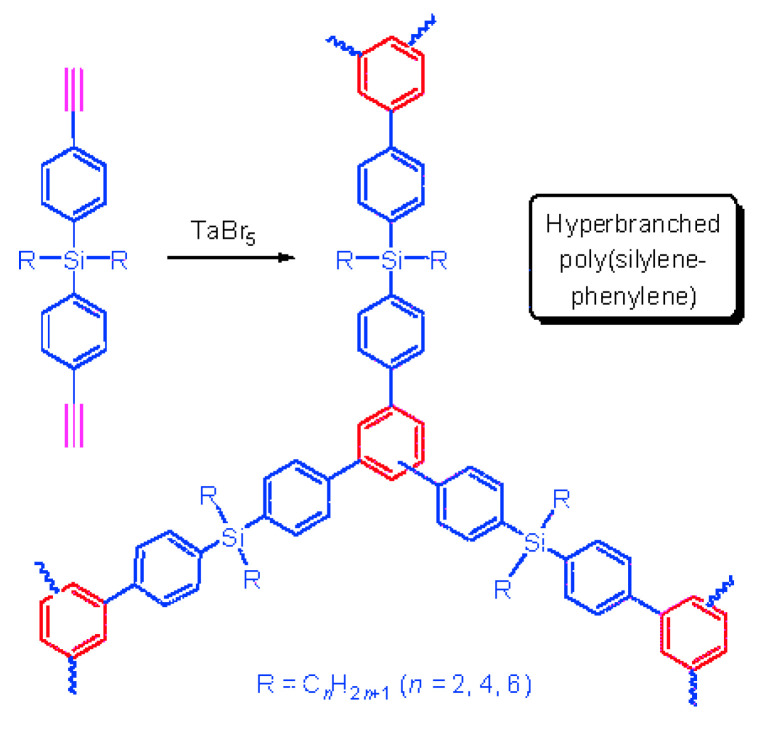
Scheme of polycyclotrimerization of sylilenediyne. Reprinted with permission from [173]. Copyright 2007, American Chemical Society.

**Figure 43 polymers-15-01105-f043:**
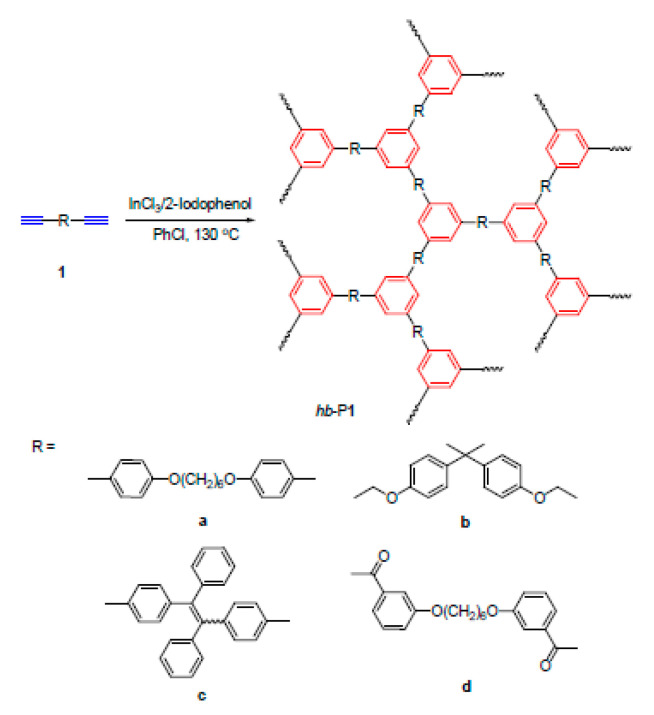
Syntheses of regioregular hyperbranched polyarylenes by InCl_3_/2-iodophenol catalyzed polycyclotrimerization of diynes. Reprinted with permission [174]. Copyright 2010 Royal Society of Chemistry (Great Britain).

**Table 1 polymers-15-01105-t001:** Influence of polymerization conditions by DEB conversion.

Sample	Solvent	T, °C	τ, min	Y, %
PDEBC-2	Benzene	82	10	59
PDEBC-4	Toluene	110	10	72
PDEBC-5	Toluene	82	120	8

**Table 2 polymers-15-01105-t002:** Poly-DEB polymerization conditions, yield, and properties (initiator n-BuLi; 55 °C; [M]_0_/[I]_0_ = 15; [M]_0_ = 0.7 mol/L) [33].

Solvent	Reaction Time τ, min	PDEB Yield, %		$M_{n}/n^{\;a}$	$H_{ol+ar}/H_{eth}{\;}^{b}$
		Soluble Fraction	Insoluble Fraction		
HMPA	3	47	-	1300/10.3	5/1
HMPA	20	48	-	1340/10.6	5/1
HMPA	40	49	-	1370/10.9	5.3/1
DMSO	1	5	-	1800/14	7/1
DMSO	20	58	23	3160/25	7/1
DMSO	60	63	26	3730/29.6	8/1

^a^ The ratio of the number of average molecular weight to the number of links in the chain; ^b^ the ratio of the sum of the integral intensities of the signals of aromatic and olefinic protons to the integral intensity of the signal of the protons of ethynyl groups.

**Table 3 polymers-15-01105-t003:** PolyDEB polymerization conditions, yield, and properties (catalyst ${\mathrm{CoCl}}_{2}\cdot 2{\mathrm{PBu}}_{3}$) [137].

Samples	Time, h	Yield, %	Solubility in THF ^a^	$${\bar{M}}_{w}$$	$${\bar{M}}_{w}/{\bar{M}}_{n}$$	$$H_{ol+ar}/H_{eth}$$
1	2	30.1	~100	10,200	2.67	5/1
2	5	53.6	90	11,500	2.85	-
3	10	73.6	80	13,300	3.20	5.1/1
4	15	75.7	80	17,500	3.31	5.6/1

^a^ At room temperature.

## Data Availability

Not applicable.

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
