# Peer review of "The Polymers of Diethynylarenes—Is Selective Polymerization at One Acetylene Bond Possible? A Review"

_polymers, 2023, doi:10.3390/polym15051105_

Round 1

Reviewer 1 Report

I commend the authors for this extensive study. My objections primarily refer to the Introduction and Conclusion.

If the editor does not mind such a (very detailed and long) Introduction, then do not take this objection into account.

I believe that the conclusion must be in accordance with the structure of the article, which is not the case now. The essential elements of the chapters of this article, 2-5, must be seen while reading the conclusion.

Some parts of the text should also be adjusted:

- 169         ...range of 77-230°K to stud.....       (230 °K)

- 171   same

- 722   .... poly-p-DEB was 5.5·10-14......

- 741 .... The sensitivity at 22 and 30 °C was -0.4 and -0.36 kHz/RH%,....

Wfat is sensitivity in kHz/RH%  ??????, same for line 750, what is "The frequency response", Additional explanations are needed, are they frequency dependencies of some AC conduction components?

Author Response

Thank you very much for your kind words and attention paid to our review. We took into account all the comments of the reviewers and introduced the necessary corrections.

The authors fully agree with the reviewer's remark about the Conclusion. The conclusion has been revised taking into account the main elements of chapters 2-5 of the manuscript and, in a modified form, is included in the manuscript.

Typos in lines 169, 171, and 722 have been corrected.

741 We agree with your comment and have made the necessary clarifications. The authors [103] used a multilayer structure consisting of layers of a conductive polymer (PDEB) on a piezoelectric substrate. This configuration has the property of a linear change in electrical impedance in response to changes in ambient temperature or humidity, which leads to a change in the frequency of the signal transmitted through the sensor. A signal with a frequency of 138 MHz was applied to the sensor as a delay line and the frequency change was measured depending on temperature and humidity.

Reviewer 2 Report

Abstract should attract reader's interest and provide an overview of the review and not so much to detail how the paper was drafted - I would suggest to completely revise the abstract

authors are using very poor language - they need to read the manuscript many times, re-phrase many parts of it and use more scientific language

not enough scientific data to confirm statements/conclusions

incomplete science

often it seems like the authors attack and claim that other articles/papers contain false data/conclusions without giving their perspective/view

many schemes where reactions are given contain typos and the authors did not even bother to make all reactions the same using the same font, size etc and/or using a software to replicate reactions from old papers

the reader gets the impressions that the authors of this review many times attack authors of published work without any apparent reason

Author Response

The authors do not consider the abstract to be detailed. It gives a general idea of the review in a concise form: the necessary information is given on certain specific aspects of the polymerization of diethynylarenes. Abstracts of numerous other reviews devoted to various aspects of the polymerization of acetylenes and having approximately the same volume [Chen, H.; Kong, J. Hyperbranched Polymers from A2 + B3 Strategy: Recent Advances in Description and Control of Fine Topology. Polym. Chem. 2016, 7, 3643–3663, doi:10.1039/C6PY00409A. Huang, D.; Liu, Y.; Qin, A.; Tang, B.Z. Recent Advances in Alkyne-Based Click Polymerizations. Polym. Chem. 2018, 9, 2853–2867, doi:10.1039/C7PY02047C. Liu, Y.; Qin, A.; Tang, B.Z. Polymerizations Based on Triple-Bond Building Blocks. Prog. Polym. Sci. 2018, 78, 92–138, doi:10.1016/j.progpolymsci.2017.09.004]. Moreover, our abstract specifically indicates which topics will not be discussed in the review. All this is important because the very large amount of information in the field of polymer chemistry should not force scientists to search for articles with information they do not need.

The authors of the review ask for examples of poor English, although they do not exclude the accidental occurrence of individual errors or typos.

Reviews should not contain any scientific data that requires confirmation, unlike scientific articles on specific studies. The purpose of the reviews is to provide complete information on any selected topical issue. Our review is written on this principle. Nevertheless, we have redone the conclusion, shortening and concretizing the text taking into account the main elements of chapters 2-5 of the manuscript. The article does not contain the results of experiments and, of course, their discussion, since this article is a review. However, in order to create an objective picture in the field of polyarylenes, in some cases we conducted a careful analysis of the cited publications. For example, the integral 1H-NMR spectrum of poly-p-DEB was analyzed in [135], and a comparative assessment of information about the intramolecular structure of polymers synthesized in [135] with the structure of these polymers synthesized under similar conditions by other authors [66, 68, 69] was carried out.

The authors disagree with this statement. On the contrary, the authors have repeatedly questioned some aspects of the articles under discussion, based on the results of a thorough analysis of the content of these articles. For example, we drew readers' attention to the incorrect interpretation of the intramolecular structure of poly-p-DEB, available in [103], the authors of which interpreted it as linear and unbranched. On the contrary, it was proved much earlier that this polymer, synthesized under identical conditions, has a branched structure [95]. The second example: we substantiated the reason for the error we found in the reaction scheme given in [65]. The third example: we drew attention to the contradictory interpretation of the intramolecular structure of poly-p-DEB synthesized using the Ni(C5H7O2)2•Ph3P complex by various groups of researchers. According to one group, polycyclotrimers were formed. According to another group, the weakly branched polyene was formed.

The authors ask to specify “many reaction schemes containing typos” to correct them. We consider it necessary to emphasize that the review uses diagrams and drawings in their original form only from literary sources to which references are given. This is done in order to exclude the possibility of accusing the authors of falsifying some other people's results. Moreover, we found and pointed out an incorrect image of the chain growth scheme during solid-phase polymerization of p-DEB in Fig. 4.

The authors are asked to indicate the sections in which they “attack the authors many times for no apparent reason” and indicate the essence of the weapons of these “attacks”. The review should not be a system of headings, followed by a simple listing of the relevant works considered, possibly with a small disclosure of the content of these works. In our review, we critically examined various aspects of the polymerization of diethynylarilenes, their structure and some properties. It is clear that the choice of the method of polymerization of diethynylarilenes affects the structure of the resulting polymers. In turn, the type of structure will affect the properties of polymers and the areas of their practical application. All this information is what readers need.

However, if there are disagreements between different authors in the interpretation of some results, it is necessary to pay attention to this. Only in this case, it is possible to help the reader find and understand the necessary information faster. This required us to make a comparative critical analysis of concrete results. It's ridiculous to talk about any attacks.

Reviewer 3 Report

This paper well organized concerning substituted polyacetylenes. The paper introduced and explained from history to recent topics. This is a very important review paper in polyacetylene chemistry. I recommend publication in the present form. 

Author Response

Thank you very much for your kind words and attention paid to our review. We took into account all the comments of the reviewers and introduced the necessary corrections. We hope that our review will be useful to you.

Round 2

Reviewer 2 Report

N/A